# *Plasmodium falciparum* transcription in different clinical presentations of malaria associates with circulation time of infected erythrocytes

Richard Thomson-Luque[1], Lasse Votborg-Novél [1,2], Wanangwa Ndovie[3], Carolina M. Andrade[1], Moussa Niangaly[2,4], Charalampos Attipa[3,5], Nathalia F. Lima[1], Drissa Coulibaly[4], Didier Doumtabe[4], Bouréima Guindo[4], Bourama Tangara[4], Fayçal Maiga[4], Abdoulaye Kassoum Kone[4], Karim Traore[4], Kassoum Kayentao[4], Aissata Ongoiba[4], Safiatou Doumbo[4], Mahamadou A. Thera [4], Boubacar Traoré[4], Karl Seydel[6,7], Nuno S. Osório [8,9] & Silvia Portugal [1,2✉]

Following *Plasmodium falciparum* infection, individuals can remain asymptomatic, present with mild fever in uncomplicated malaria cases, or show one or more severe malaria symptoms. Several studies have investigated associations between parasite transcription and clinical severity, but no broad conclusions have yet been drawn. Here, we apply a series of bioinformatic approaches based on *P. falciparum*'s tightly regulated transcriptional pattern during its ~48-hour intraerythrocytic developmental cycle (IDC) to publicly available transcriptomes of parasites obtained from malaria cases of differing clinical severity across multiple studies. Our analysis shows that within each IDC, the circulation time of infected erythrocytes without sequestering to endothelial cells decreases with increasing parasitaemia or disease severity. Accordingly, we find that the size of circulating infected erythrocytes is inversely related to parasite density and disease severity. We propose that enhanced adhesiveness of infected erythrocytes leads to a rapid increase in parasite burden, promoting higher parasitaemia and increased disease severity.

[1] Centre of Infectious Diseases, Parasitology, Heidelberg University Hospital, Heidelberg, Germany. [2] Max Planck Institute for Infection Biology, Berlin, Germany. [3] Malawi-Liverpool-Wellcome Trust Clinical Research Programme, Blantyre, Malawi. [4] Mali International Center of Excellence in Research, Malaria Research and Training Centre (MRTC), University of Sciences Techniques and Technologies of Bamako, Bamako, Mali. [5] Department of Tropical Disease Biology, Liverpool School of Tropical Medicine, Liverpool, UK. [6] Blantyre Malaria Project, University of Malawi College of Medicine, Blantyre, Malawi. [7] College of Osteopathic Medicine, Michigan State University, East Lansing, MI, USA. [8] Life and Health Sciences Research Institute (ICVS), School of Medicine, University of Minho, Braga, Portugal. [9] ICVS/3B's – PT Government Associate Laboratory, Braga, Portugal. ✉email: portugal@mpiib-berlin.mpg.de

Worldwide over 200 million malaria cases occur yearly, out of which ~2 million progress to severe disease, leading in 2019 to more than 400,000 deaths, mostly of African children under the age of five[1]. The major causative agent of malaria, *Plasmodium falciparum*, causes disease through continuous asexual intraerythrocytic developmental cycles (IDCs), each lasting ~48 h and producing 8–30 new parasites[2]. Circulation of young parasite forms within each IDC, called ring stages, is a hallmark of *P. falciparum* malaria[3], while more developed stages express parasite antigens at the host cell surface to promote their adhesion to the vascular endothelium and avoid splenic clearance[4,5]. *P. falciparum* infection can produce a range of clinical outcomes: remaining asymptomatic despite infection, presenting with fever and other non-specific symptoms in uncomplicated malaria, or exhibiting one or several signs of severe disease leading to cerebral malaria, severe anaemia, coma, pulmonary oedema, or metabolic acidosis[6]. This array of malaria presentations associates with parasite burden, and with different stages of a progressively developing protective immunity which increases with parasite exposure and host age[7,8]. Protection from severe malaria is rapidly acquired after a few clinical episodes in areas of high transmission[9], and years of exposure result in mostly asymptomatic infections in adolescence and adulthood[8,10,11]. Accordingly, in malaria-endemic areas, parasitaemia with high parasite burden is often seen in young children with severe malaria[12]. In contrast, lower parasitaemia typifies uncomplicated malaria or asymptomatic infections in older individuals[13,14], reflecting cumulative exposure and gradual naturally acquired immunity to malaria. Nevertheless, in age-matched individuals with apparently similar parasite exposures, different clinical outcomes have been linked to parasites causing severe malaria with higher or lower parasitaemias[15], or parasites promoting non-severe vs severe malaria[16,17], without allowing for major conclusions regarding the contribution of parasite transcription to disease outcome. Recently, in a cohort study in the seasonal transmission setting of Mali, we compared parasites persisting at low parasitaemias in asymptomatic children at the end of the 6-month dry season with parasites causing uncomplicated malaria in age-matched individuals during the ensuing wet season, and found highly different transcriptional signatures. We determined that gene expression differences were strongly associated with the developmental stage of circulating *P. falciparum* within the ~48 h IDC[18]. Indeed, during each IDC, *P. falciparum* follows a tightly regulated, developmental stage-dependent[19] transcriptional pattern from merozoite invasion of erythrocytes, through the ring- and trophozoite-stages, until the multinucleated schizont releases new merozoites at the end of the IDC[20–22], and parasite transcription from patients' blood can reveal the stage composition of circulating parasites[16,17,23,24]. Our recent data showed that during the dry season, *P. falciparum*-infected erythrocytes circulate longer within the ~48 h IDC and low parasitaemia is maintained for several months through increased splenic clearance[18], hinting that previously reported transcriptomes from *P. falciparum* field isolates causing distinct clinical outcomes may also have been influenced by differences in adhesion efficiency and circulation times of parasitized erythrocytes.

In this work, we revisit ten studies published between 2007 and 2020 that report parasite transcriptional differences between distinct disease severities, parasite burdens, non-negative matrix factorization (NMF) clustering of expression profiles, or transmission intensities to bioinformatically interrogate transcriptional differences dependent on developmental stage. Our approach centered on the tight transcriptional pattern governing the ~48 h IDC[20]; and based on the simple idea that genes that are differentially upregulated in more vs less-developed parasite stages comprise transcripts peaking at later stages of the IDC, whereas genes that are differentially upregulated in less-developed stages represent transcripts peaking earlier in the IDC. Accordingly, an approach using in vitro grown parasite lines previously assigned ~4400 *P. falciparum* transcripts to a particular IDC stage, allowing predictions of the stage composition of mixed parasite samples[24], which we also applied here. Furthermore, we use previously described algorithms integrating the parasites' full transcriptome to determine expected stage-compositions[17], and the maximum likelihood estimate of average parasite age in hours post erythrocyte invasion (hpi)[16] of a given sample. Applying these bioinformatic methods to previously published studies we determine the expression profile of differentially expressed genes (DEGs)[20], and predict the developmental stages present in each sample[24]. Our re-interpretation of the transcriptional analyses and the calculations of parasites' age in the various studies reveals new transcriptional signatures supporting the hypothesis that shorter circulation time of infected erythrocytes within the ~48 h IDC associates with increased parasite growth rates, virulence, and disease severity.

## Results

**Similarities and differences in *P. falciparum* transcriptional profiles across studies.** To investigate a possible broad signature of *P. falciparum* transcription associating with parasite burden and/or disease severity, we identified ten published studies between 2007 and 2020 that reported transcriptional differences between malaria parasites isolated from individuals presenting with a spectrum of malaria symptoms and parasitaemias and residing in areas of different transmission intensities (Table 1). With the exception of one study that compared acute vs chronic *P. coatneyi* infection in non-human primates in the laboratory setting[25], all studies analyzed *P. falciparum* samples from naturally infected humans. Seven studies reported transcriptional differences between *P. falciparum* samples collected from individuals presenting with distinct disease severities, parasite burdens, transmission intensities, or NMF-generated clusters immediately after blood draw[15,17,18,26–29], and two studies compared cultured late trophozoites or schizont stages of *P. falciparum* samples from individuals with different disease severities and parasitaemias[16,30]. The studies used different numbers of samples collected from diverse geographic settings, employed microarray or next-generation sequencing methodologies to quantify parasite transcripts, and used various analytical approaches to define differential gene expression. A varying number of DEGs were reported comparing higher vs lower parasitaemias[15,16,18,25,27,30], severe vs mild malaria[16,17,27,29], and mild malaria vs asymptomatic parasite carriage[18,25,30], clusters of transcriptionally similar groups of samples[15,26], or differing transmission intensities[28] (Table 1). Considering the seven *P. falciparum* studies that reported DEGs, we observed only limited and not consistent overlap of DEGs; there was not a single DEG that was common across the seven studies (Fig. 1a and Supplementary Table 1). Moreover, there was no enrichment of common terms identified across studies within gene ontology (GO) or functional category (Fig. 1b). We noted, however, the overlap between DEGs found in lower asymptomatic parasitaemias in the dry season vs higher parasitaemias in clinical malaria cases in the wet season in Mali[18], and those identified in lower vs higher parasitaemias in cerebral malaria cases in Malawi[15], and some trend of DEGs connect with the similar directionality between the two studies. Transcripts increased in higher parasitaemias in Andrade et al. were also increased in higher parasitaemias in Milner et al.. We also observed connections between upregulated transcripts in higher parasitaemias in Milner et al. and Lee et al.,

**Table 1 Characteristics of the studies included in the analyses.**

| Study | Ref. | Study site | No. of samples | Variable | Groups | Technique | DEG method | DEGs (n) |
|---|---|---|---|---|---|---|---|---|
| Almelli et al. 2014 | 30 | Cameroon | 214 | Severity/parasitemia | Asymptomatic vs cerebral malaria | Microarray | Limma package | 234 |
| Andrade et al. 2020 | 18 | Mali | 24 | Seasonality/severity/parasitaemia | Dry asymptomatic vs Wet season mild malaria | RNAseq | DESeq2 | 1609 |
| Cordy et al. 2019 | 25 | NA (lab P. cootneyi) | 4 | Disease phase/parasitaemia | Acute vs post-subRx vs chronic | RNAseq | DESeq2 | Not reported |
| Daily et al. 2007 | 26 | Senegal and Benin | 43 | NMF clustering of expression profiles | Clusters 1 vs 2 vs 3 | Microarray | GSEA-Gene Set | Not reported |
| Lee et al. 2018 | 27 | The Gambia | 46 | Severity/parasite biomass | Severe vs uncomplicated | RNAseq | EdgeR | 236 |
| Lemieux et al. 2009 | 16 | The Gambia | 17 | Severity/parasitaemia | Mild vs severe | Microarray | not specified | 278 |
| Milner et al. 2012 | 15 | Malawi | 58 | Parasitaemia/severity (CM) | High vs low parasitemia | Microarray | GenePattern | 2774 |
| Rono et al. 2017 | 28 | Kenya and Sudan | 96 | Transmission intensity | High vs low transmission settings | Microarray | Limma package | 436 |
| Tonkin-Hill et al. 2018 | 17 | Indonesia | 44 | Severity/parasitaemia | Severe vs non-severe | RNAseq | DESeq2 | 358 |
| Yamagishi et al. 2014 | 29 | Indonesia | 116 | Severity/parasitaemia | Mild vs severe | RNAseq | Not specified | Not reported |

and between severe cases of Tonkin-Hill et al. and higher parasitaemias in Andrade et al. or Milner et al., where we also detected some connections of DEGs aligned with parasitaemia directionality (Fig. 1a and Supplementary Data 1). In addition, although the GO and functional analyses did not identify significant enrichment of common pathways between studies (Fig. 1b), similar tendencies of gene expression of transcripts belonging to the fatty acid biosynthesis pathway, which appeared significantly increased in higher parasitaemias of Andrade et al., were detected in samples of higher parasitaemias in Milner et al. and Lee et al. (Fig. 1c). Likewise, the spliceosome pathway that was shown to be significantly increased in higher parasitaemias of Milner et al. showed a similar trend of overexpression in higher parasitaemias of Lee et al. and Tonkin-Hill et al. (Fig. 1c). The RNA binding pathway distinguished between groups in Milner et al. and Lee et al. (Fig. 1b, c), but also partially in Andrade et al., and in Tonkin-Hill et al. when samples were ordered by increasing number of housekeeping gene reads (Fig. 1c). Altogether these data highlight the modest overlap of DEGs between the datasets, while hinting that differences in parasitaemia observed across the studies may produce common transcriptional signatures.

**Parasite circulation time drives transcriptional difference between asymptomatic dry season infections and mild malaria-causing parasites in the wet season.** Recently, we have shown that differences in circulation time of infected erythrocytes prior to cytoadhesion between persisting low parasitaemias at the end of the dry season in asymptomatic children and malaria-causing *P. falciparum* parasites in the wet season associated with major transcriptional differences that were linked to the developmental stage of circulating parasites[18]. Now, we quantitatively characterized those associations with a series of bioinformatic approaches centered on the tight transcriptional pattern governing *P. falciparum* during the ~48 h IDC[20]. First, we outlined the expression pattern of DEGs upregulated in the low dry season parasitaemias or in the higher malaria cases' parasitaemias along the ~48 h IDC in the reference HB3 *P. falciparum* parasite line in vitro, according to Bozdech et al.[20], and obtained very different heatmaps and associated expression curves (Fig. 2a and Supplementary Data 2). While transcripts upregulated in low dry season parasitaemias mostly showed high expression between 12 and 36 h post invasion in the reference HB3 *P. falciparum* parasite in vitro, the upregulated transcripts in the higher parasitaemias of clinical malaria cases in the wet season presented the opposite trend, higher expression between 0 and 12 h post invasion, or at very late stages after 40 h (Fig. 2a), when genes of merozoite invasion and early ring stages start to be transcribed again[20]. This agrees with the presence of more developed parasite stages in the low parasitaemias in dry season samples, and young ring stages in the higher burden malaria cases[18]. Interestingly, this gene-level metric of DEGs along the IDC in the reference HB3 *P. falciparum* in vitro was independent of the fold change in expression observed between dry season and malaria cases (Supplementary Fig. 1 and Supplementary Data 2). We then used another pre-established gene-specific metric, the time of peak expression of a transcript within the ~48 h IDC determined through a periodogram analysis[24]. Examining the time of peak expression of upregulated DEGs in the dry season (low parasitaemias), and in malaria cases (high parasitaemias), we observed later peaks (24.1 h 95% CI 23.9, 24.4) in upregulated genes in the dry season, and earlier peaks (12.4 h 95% CI 11.1, 15.5) for the upregulated genes in malaria cases ($p < 0.0001$, Mann–Whitney) (Fig. 2b and Supplementary Data 2). The DEGs between low parasitaemias in the dry season and higher parasitaemias in the wet season were also assessed with another approach proposed by Painter et al.[24], which previously assigned a particular IDC stage to ~4400 *P. falciparum* transcripts, thus allowing for the prediction of stage composition of

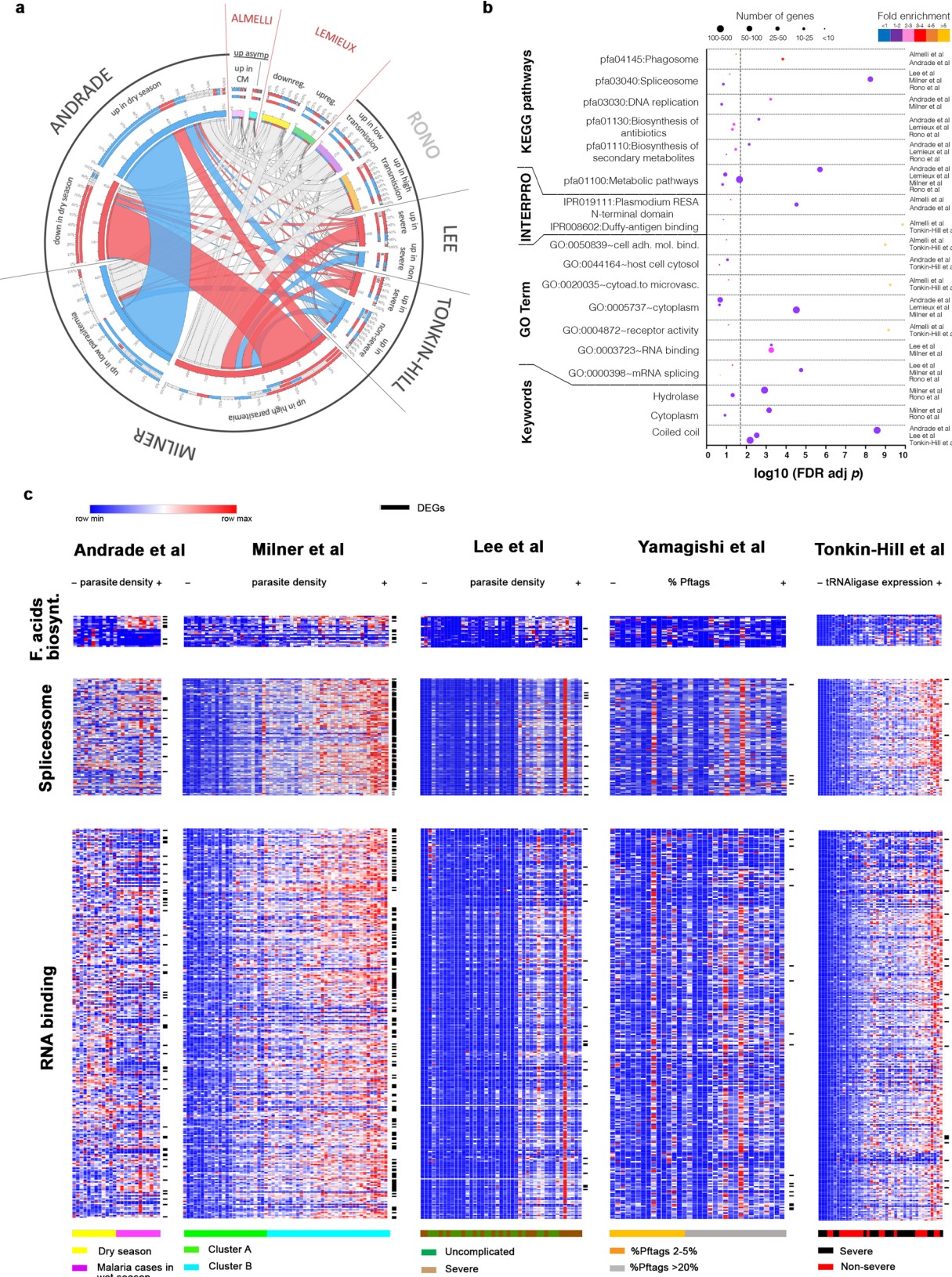

mixed parasite samples from their DEGs. This categorization applied to the upregulated DEGs in the dry (low parasitaemias) or wet seasons (high parasitaemias) again revealed more developed parasites alongside fewer ring stages from transcripts upregulated in the dry season, while transcripts upregulated in malaria cases associated with higher proportions of early ring stages ($p < 0.001$, Fisher's exact test)

(Fig. 2c and Supplementary Data 2). Also estimating the proportion of developmental stages composing each sample of Andrade et al., but this time using their entire transcriptional profile through a mixture model proposed by Tonkin-Hill et al.[17], we obtained further evidence that low parasitaemia samples in the dry season contained *P. falciparum* trophozoites, while malaria cases samples in the wet

**Fig. 1 Similarities and differences in *P. falciparum* transcriptional profiles across different studies. a** Circos plot showing interstudy connection of reported DEGs samples of Andrade et al., Milner et al., Tonkin-Hill et al., Lee et al., and Rono et al. (non-cultured), and Lemieux et al. and Almelli et al. (in vitro cultured to late trophozoites and schizont stages). From outside to inside are (i) study, (ii) variables analyzed, (iii) percentage of shared genes between modules; (iv) number of shared genes between modules. Blue highlights upregulated DEGs in lower parasitaemias; red highlights upregulated DEGs in higher parasitaemias; gray highlights DEGs from variables of studies not parasitaemia related (different colors—pink, light blue, yellow, green, purple, orange—distinguish the different variables studied). **b** Summary of GO terms, keywords, KEGG pathways, INTERPRO protein domains significantly enriched with DEGs reported in the different studies represented and sorted by adjusted $p$ value (–$\log_{10}$ Fisher's exact test one-sided $p$ values corrected for multiple comparisons using false discovery rate). The dot size indicates the number of DEGs associated with the functional group in each study and the dot color indicates the fold enrichment. Gray dashed line indicates the $p$ value threshold (–$\log_{10}$ (0.05) = 1.3). **c** Heatmaps showing normalized reads of genes involved in different GO terms (rows) for each subject (columns) of Andrade et al., Milner et al., Lee et al. Yamagishi et al., and Tonkin-Hill et al.; genes are ordered by peak time of transcription in vitro according to Painter et al.[24], Within each study, subjects are ordered by parasite densities from lowest (−) to highest (+) (in parasite/µL, % of Pf tags, or housekeeping gene expression). DEGs show "−" marks and are highlighted in Supplementary Data 1 of the source data file.

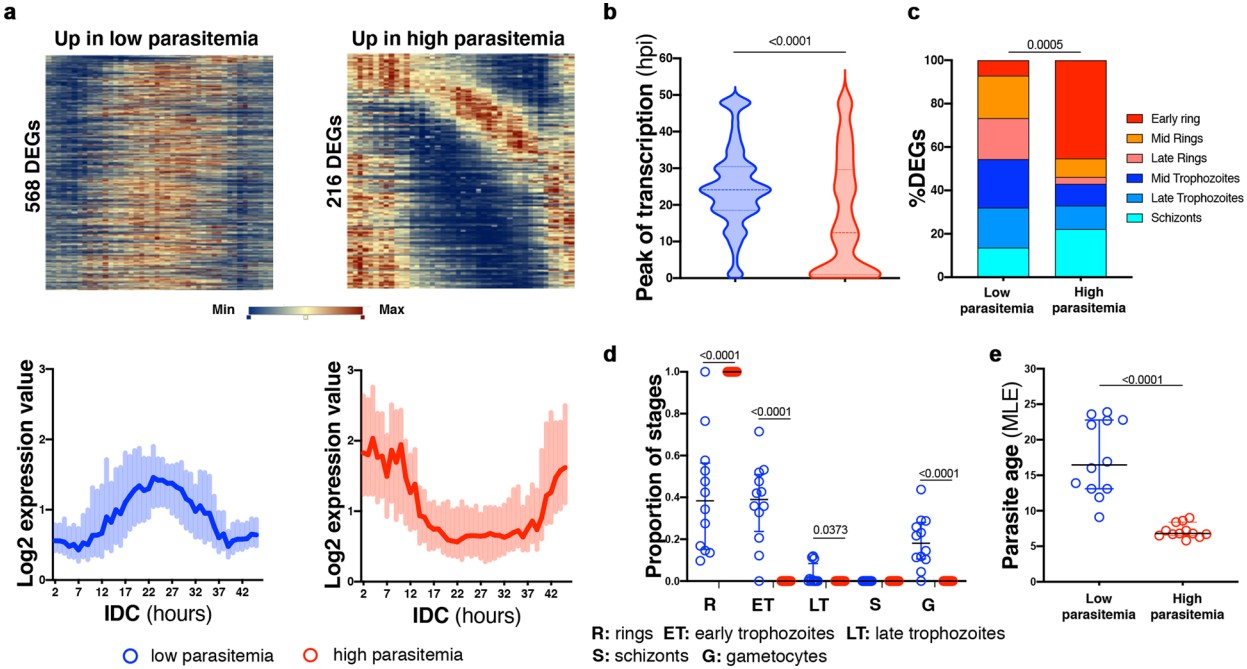

**Fig. 2 Parasite circulation time drives transcriptional differences between asymptomatic dry season infections and mild malaria-causing parasites in the wet season. a** Heatmaps (top) and $\log_2$ expression value curves (bottom) in the *P. falciparum* HB3 reference along the ~48 h IDC in vitro according to Bozdech et al.[20] of Andrade et al. DEGs upregulated in low parasitaemias at the end of the dry season (left, $n = 568$, 50.22% of the total 1131 DEGs) and upregulated in high parasitaemias in malaria cases in the wet season (right, $n = 216$, 45.38% of the total 476 DEGs). For $\log_2$ expression value curves (bottom) data indicate median ± IQR. **b** Time of peak transcription in hours post invasion in *P. falciparum* HB3 reference along the ~48 h IDC in vitro according to Painter et al.[24] of DEGs upregulated in low parasitaemias at the end of the dry season ($n = 971$, 85.85% of the total 1131 DEGs), and DEGs upregulated in high parasitaemias in malaria cases in the wet season ($n = 397$, 83.40% of the total 476 DEGs). Data indicate median ± IQR; two-tailed Mann–Whitney test, $p < 0.0001$. **c** Predicted stage composition according to Painter et al.[24] of DEGs upregulated in low parasitaemias at the end of the dry season ($n = 857$, 75.77% of the total 1131 DEGs) and DEGs upregulated in high parasitaemias in malaria cases in the wet season ($n = 221$, 46.43% of the total 476 DEGs). Two-tailed Fisher's exact test of DEGs assigned to developmental stages. **d** Predicted proportion of ring, early trophozoite, late trophozoite, schizont, and gametocyte stages according to Tonkin-Hill et al.'s mixture model[17] using whole transcriptome data from low parasitemia samples from the dry season (blue, $n = 12$) and high parasitemia samples from malaria cases in the wet season (red, $n = 12$), One-way ANOVA with Sidak multiple comparisons test. **e** Maximum likelihood estimation (MLE) according to Lemieux et al.[16] of parasite age in hours post invasion of low parasitemia samples from the dry season (blue, $n = 12$) and high parasitemia samples from malaria cases in the wet season (red, $n = 12$). Data indicate mean ± SD; two-tailed Mann–Whitney test, $p < 0.0001$.

season revealed only young ring stages and not further developed parasites ($p < 0.0001$, one-way ANOVA) (Fig. 2d and Supplementary Data 2). Finally, as reported earlier[18], the maximum likelihood estimation (MLE) method defined by Lemieux et al.[16] applied to the transcriptomes of parasites collected in the dry season and malaria cases determined that the expected age in hpi was higher in parasites circulating in the dry vs the wet season ($p < 0.0001$, Mann–Whitney) (Fig. 2e and Supplementary Data 2). Altogether, these analyses using DEGs and the whole transcript data of low vs high parasitaemias

highlight the strong link between the transcriptional signature observed in low parasitaemias and the increased age of the circulating parasites.

**DEGs from earlier studies show increased parasite circulation time in low vs high parasitaemias.** To determine if parasite circulation time could also associate with transcriptional differences identified in other studies, we applied these same

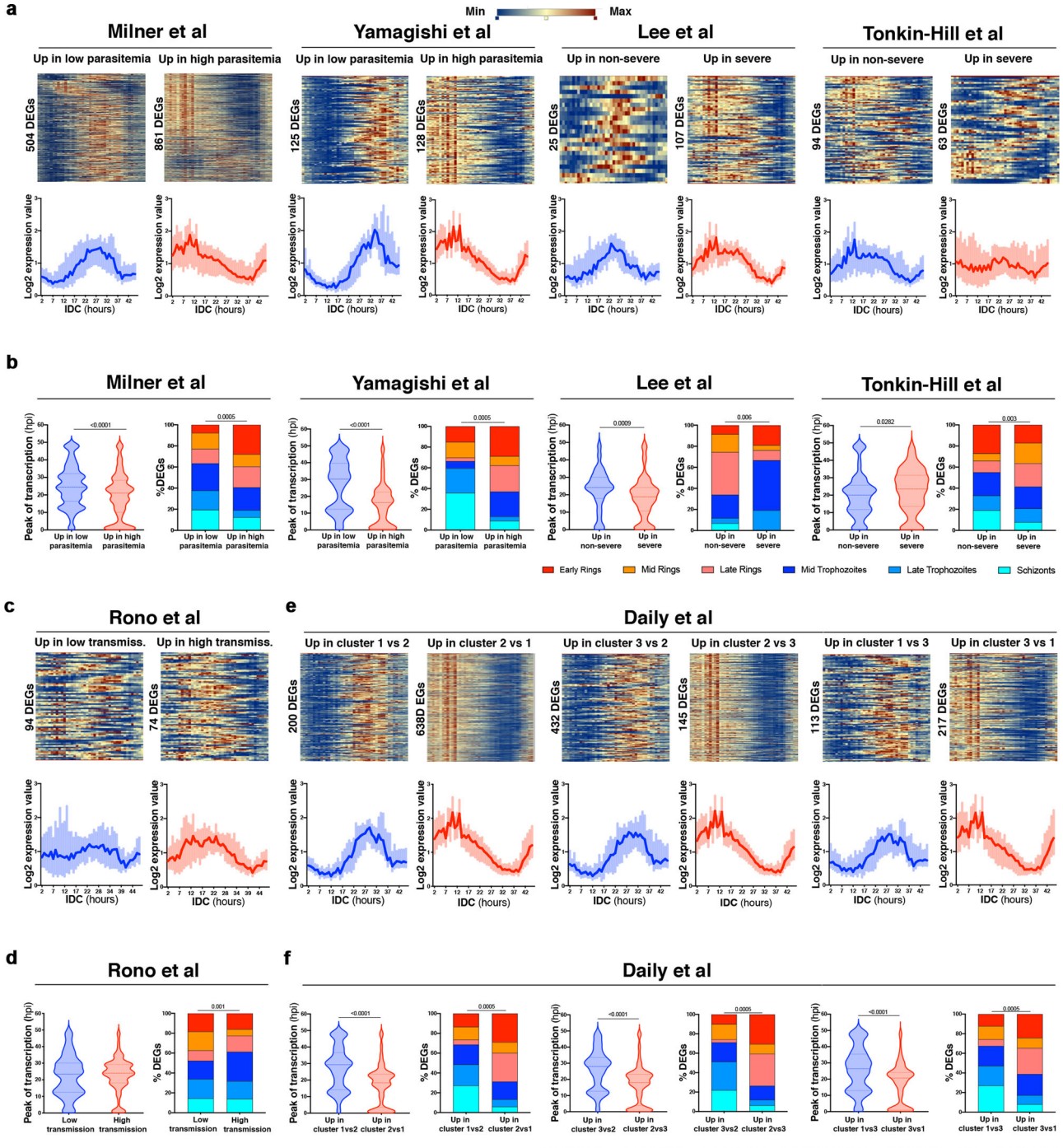

bioinformatic strategies to publicly available datasets. Some studies provided a DEG list (Milner et al., Tonkin-Hill et al., Lee et al., Lemieux et al., and Rono et al.) and for others we calculated from their accessible gene count matrices (Yamagishi et al., Daily et al., and Cordy et al., see Methods). Strikingly, we observed that the four studies comparing freshly obtained parasites from low vs high parasitaemias (Milner et al. and Yamagishi et al.) and mild vs severe disease (Lee et al. and Tonkin-Hill et al.) showed similar patterns as the observed comparison between dry and wet season parasites. Curves of expression and heatmaps along the ~48 h IDC in the reference HB3 *P. falciparum* parasite in vitro showed that DEGs upregulated in low parasitaemias or mild symptoms peak later in the IDC than the upregulated transcripts in high parasitaemias or

severe malaria cases, which mostly peak expression early post invasion in Milner et al., Yamagishi et al., and Lee et al., and to a lesser extent in Tonkin-Hill et al. (Fig. 3a and Supplementary Data 3). In agreement with these differences, the peak expression time within the ~48 h IDC according to Painter et al.[24] of upregulated DEGs in low parasitaemia samples was later in Milner et al. (25.1 h 95% CI 24.3, 25.9 in low parasitaemias, and 19.5 h 95% CI 18.8, 20.2 in high parasitaemias), Yamagishi et al. (27.1 h 95% CI 25.4, 28.9 in low parasitaemias, and 15.0 h 95% CI 13.4, 16.6 in high parasitaemias), and Lee et al. (23.2 h 95% CI 20.1, 26.3 in non-severe, and 18.6 h 95% CI 16.7, 20.5 in severe cases) ($p < 0.001$ Mann–Whitney) (Fig. 3b and Supplementary Data 3). However, in the Tonkin-Hill et al. study such difference was not found in the direction of later peak in lower

**Fig. 3 DEGs from earlier studies show increased parasite circulation time in low vs high parasitaemias. a** Heatmaps (top) and log$_2$ expression value curves (bottom) in *P. falciparum* HB3 reference along the 48 h IDC in vitro according to Bozdech et al.[20] of DEGs upregulated in low parasitaemia samples in Milner et al. and Yamagishi et al., and non-severe samples in Lee et al. and Tonkin-Hill et al.; and DEGs upregulated in high parasitaemias or severe malaria cases of the same studies. For log$_2$ expression value curves (bottom) data indicates median ± IQR. **b** Time of peak transcription in *P. falciparum* HB3 reference along the 48 h IDC in vitro (left) and the predicted stage composition (right) according to Painter et al.[24] of DEGs upregulated in low parasitaemia samples or in non-severe samples in Milner et al. (*n* = 58), Yamagishi et al. (*n* = 36), Lee et al. (*n* = 46), and Tonkin-Hill et al. (*n* = 44), and DEGs upregulated in high parasitaemias or severe malaria cases of the same studies. **c** Heatmaps (top) and log$_2$ expression value curves (bottom) in *P. falciparum* HB3 reference along the 48 h IDC in vitro according to Bozdech et al.[20] of DEGs upregulated in samples from a low transmission area, and upregulated in samples from a high transmission area in Rono et al. For log$_2$ expression value curves (bottom) data indicates median ± IQR. **d** Predicted stage composition according to Painter et al.[24] of DEGs upregulated in samples from a low transmission area and DEGs and upregulated in samples from a high transmission area in Rono et al. (*n* = 96). **e** Heatmaps (top) and log$_2$ expression value curves (bottom) in *P. falciparum* HB3 reference along the 48 h IDC in vitro according to Bozdech et al.[20] of DEGs upregulated in each of the cluster comparisons described by Daily et al. **f** Time of peak transcription in *P. falciparum* HB3 reference along the 48 h IDC in vitro (left) and the predicted stage composition (right) according to Painter et al.[24] of DEGs upregulated in each of the cluster comparisons described by Daily et al. (*n* = 43. For log$_2$ expression value curves (bottom) data indicates median ± IQR. Data in **b**, **d**, and **f** indicate median ± IQR; two-tailed Mann–Whitney test (left), and two-tailed exact Fisher's done over absolute values of DEGs assigned to developmental stages (right).

parasitaemias, nor were the differences as pronounced (20.3 h 95% CI 18.3, 22.2 in non-severe, and 22.5 h 95% CI 20.3, 24.8 in severe malaria) (Fig. 3b), likely due to the small number of DEGs found in the study and also possibly due to the largely overlapping parasitaemias between individuals included in non-severe and severe cases, which were not significantly different. The stage composition prediction according to Painter et al.[24] identified more developed parasites from upregulated transcripts in low parasitaemia groups across the different studies, while transcripts upregulated in the high parasitaemia groups associated with higher proportions of ring stages (*p* < 0.001, Fisher's exact test) (Fig. 3b); and again the comparison between non-severe and severe cases without strong parasitaemia differences included in Lee et al. or Tonkin-Hill et al. resulted in much less pronounced differences in predicted stages (Fig. 3b).

Rono et al., a study that did not segregate sample groups based on parasitaemia or severity of disease but instead compared samples from high vs low transmission settings across all parasite burdens, did not show pronounced differences in heatmaps or expression curves of DEGs upregulated in low and high transmission settings along the ~48 h IDC in HB3 *P. falciparum* reference in vitro (Fig. 3c and Supplementary Data 3), nor were the peak expression times according to Painter et al.[24] of upregulated DEGs in low or high transmission settings different (23.4 h 95% CI 18.1, 24.5 in low transmission, and 24.0 h 95% CI 23.2, 25.1 in high transmission) (Fig. 3c). Accordingly, the predicted differences of developmental stages associating with the DEGs upregulated in high or low transmission settings although significant were not particularly striking (Fig. 3d and Supplementary Data 3).

Interestingly, in Daily et al., a study clustering samples from a range of clinical presentations through NMF according to their transcriptional similarity into three groups of samples without significant differences in parasite densities, our analyses nevertheless highlighted the presence of more developed parasites in samples of cluster 1 and cluster 3 than in cluster 2. Heatmaps and expression curves along the ~48 h IDC showed later expression in DEGs upregulated in cluster 1 and 3 vs cluster 2, and the same was true for cluster 1 vs cluster 3, with DEGs upregulated in cluster 3 showing high expression in early hours of the IDC (Fig. 3e). Also, the peak expression times within the IDC according to Painter et al.[24] revealed that upregulated DEGs in clusters 1 or 3 peaked later than DEGs upregulated in cluster 2 (29.3 h 95% CI 27.0, 29.9 in cluster 1 vs 18.35 h 9% CI 17.1, 19.9 in cluster 2) and (28.0 h 95% CI 25.2, 25.9 in cluster 3 vs 18.0 h 95% CI 17.56, 19.5 in cluster 2) and also upregulated DEGs in cluster 1 peaked later than upregulated DEGs in cluster 3 (26.4 h

95% CI 23.8, 29.4 in cluster 1 and 20.6 h 95% CI 18.0, 22.2 in cluster 3) (*p* < 0.0001, Mann–Whitney) (Fig. 3f and Supplementary Table 3). This categorization is in line with what Daily et al. refer in their original study, namely, that cluster 2 is the closest to early ring-stage profiles and also the cluster with the highest average parasitaemia and thus likely younger in hpi than the other two clusters, but also that cluster 3 is the one with most severe cases and presenting higher parasitaemias than what is observed in cluster 1[26] is also in line with the predicted stage composition with later forms in the cluster 1 and 3 vs 2, and cluster 1 vs 3 (Fig. 3f).

**Whole transcriptome analyses across different studies support increased circulation time of infected erythrocytes in low vs high parasitaemias.** We applied the mixture model proposed by Tonkin-Hill et al.[17] to the whole transcriptomes of the different studies and consistently found lower proportions of rings and/or higher proportions of early trophozoites or later developmental stages in the samples originating from lower parasitaemia conditions (Milner et al. and Yamagishi et al.) and mild malaria cases (Lee et al. and Tonkin-Hill et al.). Conversely, more young stages were predicted in the samples originating from high parasitaemias or severe malaria cases in the same studies (*p* < 0.05 one-way ANOVA) (Fig. 4a and Supplementary Data S4). The MLE method by Lemieux et al.[16] applied to the four transcriptomes (Milner et al., Yamagishi et al., Lee et al., and Tonkin-Hill et al.) determined significantly more mature parasites in the low parasitaemias in Milner et al. and in the less severe malaria samples from Tonkin-Hill et al. (Fig. 4b and Supplementary Data 4), while the differences were not statistically significant in Yamagishi et al. or Lee et al. Microarray data from Daily et al. and Rono et al. were not compatible with running the mixture model proposed by Tonkin-Hill et al.[17] or the MLE calculation method defined by Lemieux et al.[16]

The particular case of the study by Cordy et al. analyzing *P. coatneyi* in four rhesus macaques is the only study involving longitudinal parasite sampling, including acute malaria with high parasitaemias and chronic low-level infection following sub-curative treatment[25]. Applying the MLE method to the whole transcriptome of the different groups of samples, we observed that a higher proportion of younger parasites were present in acute infection/higher parasitaemias than in low parasitaemias immediately or long after the sub-curative treatment (Fig. 4c and Supplementary Data 4), and that the estimated parasite age in hpi increased as the macaques' parasitaemias decreased (Fig. 4d and Supplementary Data 4).

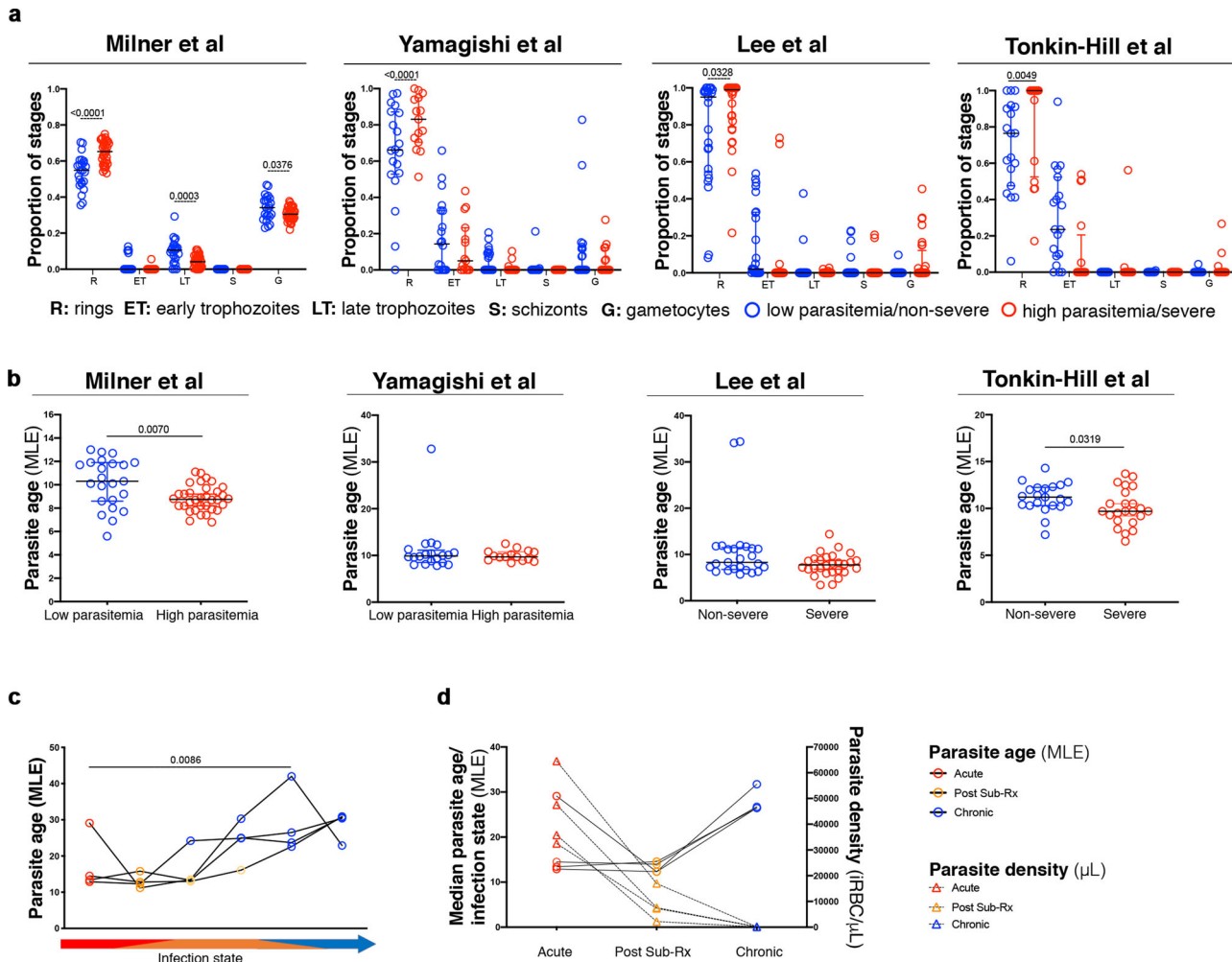

**Fig. 4 Whole transcriptome analyses across different studies supports increased circulation time in lower vs higher parasitaemias. a** Predicted proportion of ring, early trophozoite, late trophozoite, schizont, and gametocyte stages according to Tonkin-Hill et al.'s mixture model[17] using whole transcriptome data from low parasitemia samples in Milner et al. ($n = 58$). and Yamagishi et al. ($n = 36$), and non-severe samples in Lee et al. ($n = 46$) and Tonkin-Hill et al. ($n = 44$) (blue) and higher parasitaemias or severe malaria cases of the same studies (red). One-way ANOVA with Sidak multiple comparisons test. **b** Maximum likelihood estimation (MLE) of the hours post invasion of parasites in low parasitemia samples in Milner et al. ($n = 58$) and Yamagishi et al. ($n = 46$) and non-severe samples in Lee et al. ($n = 46$) Tonkin-Hill et al. ($n = 44$) (blue) and higher parasitaemias or severe malaria cases of the same studies (red). Data indicate mean ± SD; two-tailed Mann–Whitney test. **c** Maximum likelihood estimation (MLE) according to Lemieux et al.[16] of the hours post invasion of longitudinal *P. coatneyi* parasites during acute (red), post sub-curative treatment (orange), and chronic (blue) infection of four monkey samples in Cordy et al. RM one-way ANOVA (with Greenhouse–Geisser correction). **d** *P. coatneyi* parasite densities (triangles) and average parasite age by MLE (circles) during acute (red), post sub-curative treatment (orange), and chronic (blue) infection of four macaque samples in Cordy et al.

**Low parasitaemia is associated with longer circulation time of infected erythrocytes.** We then aimed to determine if parasite developmental stages in circulation correlated with parasitaemia. Some but not all studies included in the current analyses originally reported individual parasite densities (in parasites/µL of blood), which allowed us to seek associations between parasitaemia and the predicted proportions of non-ring stages according to the mixture model by Tonkin-Hill et al.[17] as a proxy for parasite development, or whenever possible, the calculated parasite age in hpi through the MLE algorithm by Lemieux et al.[16]. In accordance with our hypothesis of longer circulation of infected erythrocytes with more mature parasites in low parasitaemias, we found significant negative correlations between the proportion of non-rings (Fig. 5a and Supplementary Data 5) or the predicted age in hpi (Fig. 5b and Supplementary Data 5) and parasite density of samples in different studies ($r = -0.64$ and $r = -0.59$, respectively; $p < 0.0001$). We were also able to measure

the size of infected erythrocytes on thick smears collected at the time of blood draw in two of these studies (Andrade et al. and Milner et al.), and we confirmed that the average parasite size was significantly larger in low than in high parasitaemias (Fig. 5c, d and Supplementary Data 5). Of note, we illustrated with 3D7 *P. falciparum* parasites grown in vitro after merozoite isolation and synchronized invasion of erythrocytes, that differences in parasite sizes at early hours of development can be detected (Supplementary Fig. 2). Accordingly, despite an obvious overlap close to overall median size within each study, the proportions of parasites falling into the 1st and 4th quartiles were very much linked with parasite burden; with high parasitaemias showing larger proportions of parasites in the 1st quartile and low parasitaemias showing increased proportions of 4th quartile sized parasites (Fig. 5e and Supplementary Data 5). For these two studies we could also assemble a correlation matrix including six quantified features in our analyses (circulating parasite size, estimated

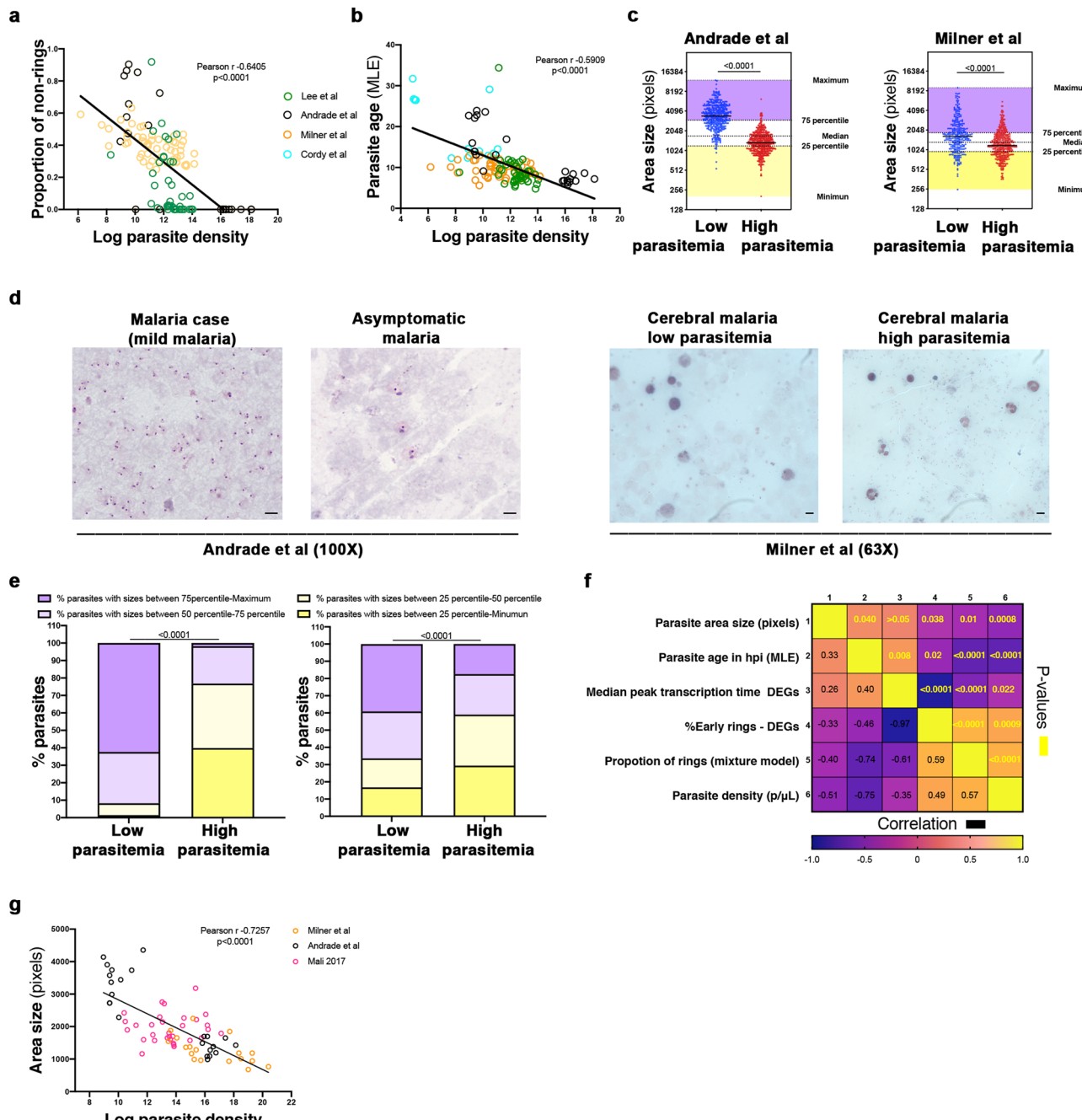

**Fig. 5 Infected erythrocytes circulate longer in low than in high parasitaemias. a** Correlation between parasite density and estimated proportion of non-ring stages according to Tonkin-Hill et al.[17] of all samples in Andrade et al. (black), Milner et al. (orange), Lee et al. (green). $p$ and $r$ determined by Pearson correlation. **b** Correlation between parasite density and average parasite age by MLE in hours post invasion estimated according to Lemieux et al.[16] of all samples in Andrade et al. (black), Milner et al. (orange), Lee et al. (green), and Cordy et al. (light blue). $p$ and $r$ determined by Pearson correlation. **c** *P. falciparum* area measured from thick smears in low parasitaemia samples (blue) in Andrade et al. and Milner et al., and in high parasitaemia samples (red) of the same studies. Yellow and purple shaded regions indicate 1st and 4th quartiles highlighting smallest and largest parasite sizes, respectively. Data indicate median ± IQR; two-tailed Mann–Whitney test, $p < 0.0001$. **d** *P. falciparum* parasites in thick blood films collected straight from the arm of children with low parasitaemia samples (left) in Andrade et al. and Milner et al., and high parasitaemia samples (right) of the same studies. Data show measurements of one of 3–4 independent readers blinded of sample grouping with interclass correlation of 0.963 and 0.889 for Malian and Malawian samples, respectively. Scale bar = 5 μm. **e** Proportion in percentages of parasites imaged in thick blood films collected directly from children with low and high parasitaemia samples in Andrade et al. ($n = 24$) and Milner et al. ($n = 32$), falling into 1st, 2nd, 3rd, and 4th quartiles. Fisher's exact done over absolute values of parasites assigned to the different quartiles. **f** Correlation matrix of variables obtained or determined from Andrade et al. and Milner et al. The cells are colored according to the scale: yellow indicates positive correlation and purple indicates negative correlation. Correlation matrix of every variable with each other without adjustment for multiple comparisons. $p$ values determined by Pearson correlation are highlighted in yellow. **g** Correlation between parasite density and parasite areas measured from thick smears in samples across a wide range of parasitaemias including Andrade et al., Milner et al., and a third study (Coulibaly et al. in preparation) with samples from uncomplicated and severe malaria cases collected in Mali. $p$ and $r$ determined by one-tailed Pearson correlation.

parasite age according to Lemieux et al.[16], median peak expression time of DEGs according to Painter et al.[24], and proportion of rings estimated by DEGs, and by the mixture model according to Painter et al.[24] and Tonkin-Hill et al.[17], respectively, and parasite density), and we observed that while the area size of circulating parasites correlated positively with estimated age in hpi and peak expression times of DEGs (all indicative of more developed parasites); it correlated negatively with the proportion of ring stages predicted by Painter categorizations of DEGs or by the full transcriptome mixture model method, and it also correlated negatively with parasite density (Fig. 5f and Supplementary Data 5). Finally, we aimed to cover a comprehensive range of parasite densities and investigate a possible correlation with size of circulating parasites. We compiled the measurements of parasites from the two studies mentioned above, Milner et al. comparing cerebral cases with high and low parasitaemias, and Andrade et al. comparing asymptomatic vs mild malaria cases (Fig. 5c, d), and we added measurements from samples from a third study (Coulibaly et al. in preparation, see Methods) comparing severe vs mild malaria. The correlation between parasite densities found in the blood of children and the parasite sizes across the studies was statistically significant and negative ($r = -0.726$, $p < 0.0001$), and particularly evident at low parasitaemias where circulating parasites are larger (Fig. 5g and Supplementary Data 5).

## Discussion

Although the parasite and host factors that determine the clinical outcome of *P. falciparum* infections are not fully understood, higher parasitaemia is frequently associated with poorer prognoses[6,15,16,18,30,31]. In this study, we accessed publicly available transcriptional profiles of parasites from reports across the malaria clinical spectrum and parasitaemia range. We showed a clear relation between circulation of older parasites within each ~48 h asexual replication cycle and lower parasitaemia. Our data support a model in which higher parasite burdens result from parasites sequestering from peripheral circulation by competently binding to endothelial cells and thus evading splenic clearance, contributing to more rapid parasitaemia growth and worse clinical presentations. Asymptomatically infected children typically present with low-level parasitaemias, which are often submicroscopic[32,33], mild, or uncomplicated malaria cases typically associated with higher parasite burdens than the ones observed in asymptomatic infected children[18,30], but lower than what is frequently seen in severe or complicated malaria cases[31]. This gradient led to the hypothesis that intrinsic replication rates of *P. falciparum* may contribute to the level of parasitaemia. However, in vitro culture of parasites collected from uncomplicated and severe malaria cases fails to consistently explain parasitaemia differences found in clinical settings[34,35], and our recently published data show similar in vitro replication ability between parasites causing mild malaria and asymptomatic infection[18]. Furthermore, we recently reported transcriptional differences between parasites causing mild and subclinical malaria associated with increased circulation time of parasites causing subclinical malaria[18], which we confirmed here (Fig. 2). Cumulative parasite exposure and the acquisition of humoral immunity are surely at the center of clinical protection in malaria-endemic areas, but the factors that determine whether similarly exposed individuals present with symptoms or not, or which parasites will increase to disease-causing levels in each individual are unknown. We propose that the binding ability to vascular endothelium of infected erythrocytes that are not cleared by antibodies affects the rate of increase of parasite load and hence malaria severity, which is supported by the ordered acquisition of antibodies recognizing

PfEMP1 on the surface of iRBCs with earlier responses observed to pathogenic domain variants with particular binding phenotypes[36,37]. In agreement with this concept, our present data reanalyzing multiple published studies comparing parasite transcription of higher vs lower parasitaemias, or more vs less severe malaria-causing parasites identified transcriptional profiles of younger parasites in circulation in higher parasitaemias and more severe malaria cases than in lower parasitaemias or more mild malaria cases (Figs. 2–4). A simple overlay of DEGs lists could not show extensive parallels between the studies, and affected molecular pathways showed only moderate similarities (Fig. 1a–c), which may be in part be explained by the many differences in methodology and analytical pathways and overlapping parasitaemias between groups of samples in some studies (Table 1). However, using analytical approaches described earlier to infer the stage composition and parasite developmental age from DEGs[20,23,24], and from whole transcriptome[16,17], we showed within each study comparing parasite of lower vs higher parasitaemias that we could detect more mature parasites in the lower parasitaemia samples and less mature signatures in the higher parasitaemia samples. Transcripts upregulated in higher parasitaemias tended to be early-expressed genes in the ~48 h IDC, while upregulated transcripts in lower parasitaemias were later-expressed genes within the asexual cycle; across the whole transcriptome through algorithms previously established, more developed parasites were consistently predicted in lower vs higher parasitaemias (Figs. 2 and 4). A study not involving parasite burden or clinical severity in the original parasite transcriptome analysis, but instead comparing *P. falciparum* from higher and lower transmission settings across all parasitaemias[28], did not reveal any signature related to parasite circulation time when we applied our analytical approach (Fig. 3c, d). None of the human parasite studies available for this reanalysis reported longitudinal data of *P. falciparum* infections, which would help determine if the progression of infection and consequent immunity would lead to a signature of more developed circulating parasites. However, a non-human primate malaria study analyzing transcription of *P. coatneyi* during acute malaria, and following sub-curative anti-malarial drug treatment that allowed immunity to control infection to low parasitaemias[25] provided us longitudinal data of high parasitaemia during symptomatic malaria and low parasitaemias in the absence of symptoms. We found that the orthologous transcriptional signature of the higher symptomatic parasite burden seen in acute malaria aligned with less-developed, younger parasites forms than the low parasite burden following sub-curative treatment (Fig. 4c, d).

In two studies included in this reanalysis, data supporting further developed parasites circulating in lower parasitaemias and less severe malaria cases were already reported at that time, albeit possibly not clearly or knowingly. Tonkin-Hill et al. described a bias toward early trophozoite transcription in the uncomplicated compared to the ring-stage transcriptional profile of the severe malaria cases analyzed ex vivo[17], and Lemieux et al. reported that a shorter period of hours in culture was needed to achieve schizont stages in parasites collected from uncomplicated cases (24 h 95% CI 23, 44) than the period needed to have parasites reach the same stage from severe malaria cases (38 h 95% CI 24, 48) ($p = 0.05$ t test)[16]. These are two clues in two different publications, of what we now put in evidence across multiple studies, and is possibly a major driver of *P. falciparum* ability to grow within a host. Early stage adhesion of infected erythrocytes results in effective evasion of splenic clearance and a subsequent rapid increase in parasitaemia and severity of disease. With two studies[15,18] we could relate the parasite transcriptional signatures with size measurements of circulating parasites within the lower and higher parasitaemia groups, which confirmed the presence of

more mature parasites in individuals with lower parasitaemias (Fig. 5d), validating longer circulation within the ~48 h IDC detectable through transcriptomic analyses leads to less rapid increase in parasite burden. Adding a third study (Coulibaly et al. in preparation) we were able to cover a wide range of parasite densities across an array of clinical presentations and quantify parasite sizes of freshly isolated parasites from multiple studies evidencing more developed parasites in samples from lower parasitaemias and milder malaria forms, and showing a negative association between parasitaemia and developmental stage of circulating parasites (Fig. 5g). It remains to be explained how *P. falciparum* alters its adhesive properties driving increased splenic clearance and maintaining lower parasitaemias. We propose that circulating antibodies control parasites with stronger binding abilities and thus potential fast-growth, leading to the biased presence in the circulation of immune or semi-immune individuals of parasites binding less efficiently. These parasites would not be able to increase to high levels because their longer presence in circulation promotes splenic clearance. Although very difficult due to the variable nature of the parasite gene families promoting cytoadhesion, the study of sequential expression and humoral responses to variant surface antigens in *P. falciparum* may inform how less-adhesive parasites appear gradually in individuals, and how virulence is regulated favoring persistence. Within the ten studies reanalyzed here, only Andrade et al. and Tonkin-Hill et al. discussed possible associations between disease severity and variant gene family transcripts, but none of the studies included longitudinal sampling. In the small number of samples analyzed in Andrade et al., there was no significant enrichment of particular *vars*, and only a trend of higher expressed *var* genes in individuals with clinical malaria vs asymptomatic in the dry season[18]. Tonkin-Hill et al. detected no differences between severe and uncomplicated malaria cases in total number of *var* gene reads, but identified segregation at the multidomain and individual domain level between severe and uncomplicated disease. However, Tonkin-Hill et al. also found genes involved in PfEMP1 transport and regulation to be downregulated in severe malaria leading the authors to suggest that *var* gene expression was reduced in severe cases[17], which in light of our current data could also simply indicate less-developed parasites.

Although we favor a hypothesis where lower parasitaemias and milder malaria cases are promoted by decreased cytoadhesion of longer circulating parasites, achieved through imposed switching following humoral immunity against better-binding parasite variants, we cannot exclude alternative scenarios. Other mechanisms affecting circulation of infected erythrocytes linked to decreased expression levels of PfEMP1[18,30], the potential effect of the host febrile temperatures[38,39] and cytokine environment[40–44], other host cues independent of humoral immunity[45] and promoted by different clinical presentations may be at place; or even the possibility that factors independent and parallel to longer circulating parasites should be investigated.

In conclusion, we revealed through transcriptional signatures that parasite circulation time associates with growth potential and parasite virulence, influencing disease outcome, and potentially highlight cytoadhesion dynamics as a major force driving the clinical prognosis of malaria.

## Methods

### Description of studies and transcriptional datasets included
The studies included in the analyses were selected through a PubMed search using "*Plasmodium*" "transcription" "field samples," and similar terms, and we further identified studies cited in the search results. Ten studies were included in the transcriptional analyses.

Andrade et al. 2020: 12 *P. falciparum* RDT+ samples from asymptomatic children in the dry season, and 12 malaria case samples from the wet season in Mali. DEGs were reported.

Milner et al. 2012: 58 *P. falciparum* cerebral malaria samples clustered into low (24) and high (34) parasitaemia in Malawi. DEGs were reported.

Yamagishi et al. 2014: 116 samples from varying *P. falciparum* parasitaemias (defined by % of Pf RNAseq tags within a sample ranging from 0.001 to 72.95%) and clinical presentations in Indonesia. DEGs were not originally reported. We determined DEGs between 16 samples with Pf tags >20% (high parasitaemia) and 20 samples with Pf tags between 2 and 5% (low parasitaemia).

Lee et al. 2018: 21 samples from uncomplicated *P. falciparum* malaria cases and 25 severe *P. falciparum* malaria samples, including cerebral malaria (5), hyperlactatemia (8), and both (12) from The Gambia. DEGs were reported.

Tonkin-Hill et al. 2018: 21 samples from non-severe *P. falciparum* malaria cases, and 23 samples from severe malaria patients in Papuan adults. DEGs were reported.

Rono et al. 2017: 96 samples of *P. falciparum* uncomplicated malaria cases from areas of different transmission intensities in Kenya and Sudan. DEGs were reported.

Daily et al. 2007: 43 samples from *P. falciparum* positive individuals across diverse clinical presentations and age clustered in three groups. DEGs were not originally reported. We determined DEGs including all samples and using the same clustering.

Lemieux et al. 2009: 8 samples from *P. falciparum* mild malaria cases and 9 samples from severe *P. falciparum* malaria in The Gambia cultured until the schizont stage. DEGs between each group and reference 3D7 were reported.

Almelli et al. 2014: 3 pooled samples of six *P. falciparum* field isolates effectively cultured in vitro until trophozoite stage from asymptomatic children, children with uncomplicated malaria, or children with cerebral malaria in Cameron. DEGs were reported.

Cordy et al. 2019: *P. coatneyi* samples collected during acute infection, following sub-curative treatment and at chronic stage of four laboratory-reared Rhesus macaques. DEGs were not reported.

An 11th study was included to investigate the relation of parasite sizes on thick blood smear and parasite density, which was not included in any gene expression analyses.

Coulibaly et al. in preparation: a case–control study of severe malaria in Bandiagara, Bamako, Sikasso, and satellites villages with 6-month to 10-year-old participants enrolled and followed from October 2014 to December 2018 in Mali. Cases were recruited among children hospitalized or seeking care with cerebral malaria (Blantyre score ≤2) or severe anaemia (hemoglobin level ≤5 g/dl) at the dedicated health facilities. Controls were children suffering from uncomplicated malaria seeking care at the same health facilities and matched by age class, residence, sex and ethnicity to the index case. The study protocol obtained ethical clearance from the Ethics Committee of Faculty of Pharmacy and Faculty of Medicine and Odonto-stomatology, University of Sciences, Techniques and Technologies of Bamako, Mali; letter of approval #2014//97/CE/FMPOS. Individual informed consent was obtained from parents or guardians. Data were anonymized to guarantee confidentiality of volunteers' identities.

### Gene expression and DEG overlap between studies
Most studies provided gene expression data; however, Lee et al. did not. Raw sequencing reads from Lee et al. and Andrade et al. were processed through Cutadapt (-q 30 --pair-filter=any -m 25) and mapped to *P. falciparum* 3D7 (PlasmoDB release 47)[46] by Hisat2[47]. Reads of Lee et al. were additionally mapped to human (UCSC hg38)[48] to only use reads that uniquely mapped to *P. falciparum*. FeatureCount (-g ID --primary -C -Q 30 -p -t gene)[49] were used to create the final gene expression table. Gene annotations were obtained from GENCODE (release 34)[50] and PlasmoDB (release 47). Conversion between new and old gene names was done by gene aliases from PlasmoDB (release 47), and genes of Cordy et al. were converted into the annotated *P. falciparum* orthologs.

DEGs of studies not providing a DEG list (Yamagishi et al., Daily et al., and Cordy et al.) were determined by the quasi-likelihood F-test in the EdgeR package[51] with TMM normalization[52], $p < 0.05$ and fold change >1.5. The EDAseq package[53] was used for principal component analysis, relative log expression (RLE), and mean variance trend. Samples in each study with mean RLE exceeding ±0.1 after limma/voom normalization[54,55] were excluded from analysis. Genes of all studies were filtered by more than 2 counts per million in ten or more samples.

Shared reported DEGs between different studies were assessed and highlighted through the online platform "circos table viewer" (http://mkweb.bcgsc.ca/tableviewer/visualize/)[56], and the Venn diagrams obtained through the platform from Bioinformatics and Evolutionary Genomics at Ghent University (http://bioinformatics.psb.ugent.be/webtools/Venn/) available online.

### Gene ontology (GO) and functional category analysis
Over-represented functional annotations, keywords, and pathways were defined using DAVID[57] for GO terms, keywords, KEGG pathways, and INTERPRO protein domains, and sorted by adjusted $p$ value (–log10 (Bonferroni-corrected $p$ values) for the different studies. The number of DEGs associated with the functional group in each study was calculated together with the fold enrichment. The statistical threshold was set as (–log10 (0.05) = 1.3.

### GO term-related heatmap
Normalized reads of DEGs involved in GO terms (rows) for each subject (columns) were included in heatmaps, where samples from

each study subject were ordered by increasing parasite density (parasite/μL) (Andrade et al., Milner et al., and Lee et al.); by %Pf tags (Yamagishi et al.); or by the expression of the *P. falciparum* housekeeping gene glycine-tRNA ligase (PF3D7_1420400) (Tonkin-Hill et al.). Within each GO group genes were ordered by their peak time of transcription in vitro according to Painter et al.[24].

**DEGs from studies expression in *P. falciparum* HB3 in vitro: heatmaps and expression curves**. For transcripts found up- or downregulated in each study comparison, we plotted the relative expression in *P. falciparum* HB3 parasite line in vitro along the 48 h IDC defined by Bozdech et al.[20], which reported microarray gene expression for 3719 *P. falciparum* genes over the 48 timepoints of the IDC. Heatmaps were generated with online application Morpheus (https://software.broadinstitute.org/morpheus/). Expression curves show the median ± IQR of the log2 expression values along the 48 h IDC.

**DEGs from different studies expression in *P. falciparum* 3D7 in vitro: peak of transcription and developmental stage categorization**. For transcripts found up- or downregulated in each study comparison, we plotted the hours post invasion at which transcription peaked in *P. falciparum* 3D7 parasite line in vitro along the IDC defined by Painter et al.[24] with a Lomb–Scargle periodogram approach as previously described[58], to define the peak time of total abundance for 5428 genes over the 48 timepoints throughout the 48 h IDC.

Painter et al.[24] further used the expression data of *P. falciparum* 3D7 parasite line in vitro along the IDC to bin transcripts based on their peak times of transcription, stabilization, and decay into groups representing six developmental stage: early ring (0–10 hpi), mid-ring (11–15 hpi), late ring/early trophozoite (16–21 hpi), mid-trophozoite (22–26 hpi), late trophozoite (27–32 hpi), and schizont (33–48 hpi). We used this categorization to show the proportion of DEGs found up- and downregulated in each study comparison.

**Whole transcriptome on a mixture model of developmental stage categorization**. The mixture model described by Tonkin-Hill et al.[17], which used RNAseq reference data of López-Barragán et al.[23], was applied to the different studies. In brief, proportions of parasite life cycle stages were estimated in a constrained linear model by fitting to transcriptomic profiles of known stages. Data with old gene names were converted and any duplicated genes were collapsed by average expression value. Genes with less than 2 counts per million in ten or more samples were excluded.

**Whole transcriptome and maximum likelihood estimation of parasite hours post invasion**. Described by Lemieux et al.[16] with microarray reference data of Bozdech et al.[20], the maximum likelihood method was applied as to the different studies transcriptome. In brief, hpi was estimated by the peak in probability of parasite age given by hourly gene expression data of the HB3 strain. Data with new gene names were converted according to the reference set and any duplicated genes were collapsed by average expression value.

**Measurement of parasite sizes and parasite density correlation**. *P. falciparum* parasites were measured on Giemsa- or Field-stained thick blood films made at the time of the blood draw for RNA extraction and/or parasite density quantification of samples from Andrade et al., Milner et al., and Coulibaly et al. in preparation. Slides were imaged to obtain 3–20 photomicrographs blindly acquired from each case with a 63× or 100× objective. Fiji software Image J was used to independently quantify the parasite areas by three to four researchers. The average measure of the interclass correlation for Malian and Malawian samples was 0.963 and 0.889, respectively. Conversion from 63× sizes to 100× was obtained using the πr2 formula. Parasite densities were personally communicated.

**Synchronized 3D7 *P. falciparum* culture**. 3D7 *P. falciparum* parasites synchronized to schizont stage were added E64 compound (Sigma-Aldrich) to prevent egress of merozoites for 6–8 h, and later merozoites were purified through filtration and cultured with non-infected RBCs and RPMI medium supplemented with Albumax-II for up to 16 h.

**Statistical analysis**. For continuous variables two-tailed Mann–Whitney or Kruskal–Wallis was used to test for differences between two or more groups respectively. Fisher's exact test was used on absolute DEG counts assigned to different developmental stages according to Painter et al. One-way ANOVA with Sidak multiple comparisons test was used to compare proportion of similar stages between groups of each study according to Tonkin-Hill et al.'s mixture model. Pearson correlation or Spearman's rank correlations were obtained accordingly. Statistical significance was defined as a two-tailed *p* value of ≤0.05. All analyses were performed with GraphPad Prism software version 8.0 or 9.0,or R (http://www.R-project.org).

**Reporting summary**. Further information on research design is available in the Nature Research Reporting Summary linked to this article.

## Data availability

The RNAseq and microarray data analyzed during the current study are available through: (1) NCBI's Gene Expression Omnibus (https://www.ncbi.nlm.nih.gov/geo/) under GEO Series accession number GSE148125 (Andrade et al. 2020), GSE83667 (Milner et al. 2012), GSE108034 (Rono et al. 2017), GSE9152 (Daily et al. 2007), and GSE103259 (Cordy et al. 2019); (2) ArrayExpress database at European Molecular Biology Laboratory European Bioinformatics Institute (www.ebi.ac.uk/arrayexpress) under accession number E-MTAB-6413 (Lee et al. 2018) and E-TABM-591 (Lemieux et al. 2009); (3) GitHub for Tonkin-Hill et al. 2018 (https://github.com/gtonkinhill/falciparum_transcriptome_manuscript/tree/master/all_ gene_analysis); and (4) Supplementary material (Supplementary Table 2) for Yamagishi et al. (2014) (https://genome.cshlp.org/content/suppl/2014/06/26/gr.158980.113.DC1.html).

DEGs were reported and accessed through Supplementary material for Andrade et al. (2020), Milner et al. (2012), Lee et al. (2018), Tonkin-Hill et al. (2018), Rono et al. (2017), Lemieux et al. (2009), and Almelli et al. (2014).

Fastq files or gene count tables were collected as stated in the Methods: Andrade et al. (2020) (https://www.ncbi.nlm.nih.gov/geo/query/acc.cgi?acc=GSE148125); Milner et al. (2012) (https://www.ncbi.nlm.nih.gov/geo/query/acc.cgi?acc=GSE83667); Yamagishi et al. (2014) (Supplementary Table 2: https://genome.cshlp.org/content/suppl/2014/06/26/gr.158980.113.DC1.html); Lee et al. (2018) (https://www.ebi.ac.uk/arrayexpress/experiments/E-MTAB-6413/samples/); Tonkin-Hill et al. (2018) (https://github.com/gtonkinhill/falciparum_transcriptome_manuscript/tree/master/all_gene_analysis/data/fc_genes_with_all_samples_aligned_human_Pf_WithOutVivax_subread_uH.RData); Rono et al. (2017) (https://www.ncbi.nlm.nih.gov/geo/query/acc.cgi?acc=GSE108034); Daily et al. (2007) (https://www.ncbi.nlm.nih.gov/geo/query/acc.cgi?acc=GSE9152); Lemieux et al. (2009) (https://www.ebi.ac.uk/arrayexpress/experiments/E-TABM-591/samples/); Cordy et al. (2019) (https://www.ncbi.nlm.nih.gov/geo/query/acc.cgi?acc=GSE103259).

DEGs were collected from Supplementary material for studies reporting it as stated in the Methods: Rono et al. (2017), Lemieux et al. (2009), and Almelli et al. (2014).

Source Data are provided with this paper.

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

## Acknowledgements

We thank the participants of all mentioned studies. We are grateful to Terrie Taylor for facilitating the collaboration with the Blantyre Malaria Project, Blantyre, Malawi. We thank Joana Martins for initial discussions on bioinformatic analyses and approaches. We thank Martin Rono and Margaret Mackinnon (KEMRI-Wellcome Trust in Kilifi, Kenya) for kindly sharing parasite densities of Rono et al. participants. We thank Peter Crompton and Tuan Tran for critically reading the manuscript. We acknowledge PlasmoDB resource for all information and regularly updating the database. This work was supported by the German Center for Infection Research (DZIF), the European Research Council under the European Union's Horizon 2020 Research and Innovation Programme (grant agreement 759534), the Deutsche Forschungsgemeinschaft (DFG, German Research Foundation) projektnummer240245660 SFB 1129 of the German Research Foundation, and the Division of Intramural Research, National Institute of Allergy and Infectious Diseases, of the National Institutes of Health. R.T.-L. and L.V.-N. were funded by SFB 1129 projektnummer240245660. R.T.-L. was funded by the European Union's Horizon 2020 Research and Innovation Programme under Marie Skłodowska-Curie grant agreement DLV-839998. The case–control study included to investigate the relation between parasite sizes on thick blood smear and parasite density was funded by the National Institute of Allergy and Infectious Diseases of the National Institutes of Health under Award Number R01AI099628 to M.A.T.

## Author contributions

R.T.-L., L.V.-N., W.N., C.M.A., M.N., C.A., and S.P. gathered and analyzed data and measured parasites sizes; D.C., D.D., B.G., B. Tangara, F.M., A.K.K., K.T., K.K., A.O., S.D., M.A.T., B. Traoré, and K.S. designed, conducted, and supervised field work generating the clinical data and samples used to measure parasite sizes; R.T.-L., L.V.-N., and N.O.S. performed bioinformatic analysis; C.M.A., M.N., N.F.L., and S.P. conducted 3D7 synchronized infection; R.T.-L. prepared figures and helped prepare the manuscript; S.P. designed the study and wrote the manuscript. All authors read and approved the final manuscript.

## Funding

## Competing interests

The authors declare no competing interests.
