## [Peer Review File · Nature Communications]

Reviewer comments, first round –

Reviewer #1 (Remarks to the Author):

NCOMMS-21-09067

Hidden transcriptional signatures of *P. falciparum* in different clinical presentations of malaria.

This manuscript by Thomson-Luque et al undertakes a detailed analysis of a number of previous studies of gene transcription in *P.falciparum* field isolates. They conclude that parasites taken from patients with low parasitaemia or non-severe disease show a pattern of transcription consistent with later stages of the asexual red cell cycle than those taken from patients with either high parasitaemia or severe disease.

The first analysis that they undertake is to compare lists of differentially expressed genes DEGs in the different categories (high/low parasitaemia, severe/non severe disease) between the previous studies. I found this to be the weakest section of the paper. The circos plot in Figure 1a shows groups of DEGs from the various studies that are concordant between the studies. In fact, the only significant concordance by eye is between the studies of Andrade et al and Milner et al. The studies of Lee et al and Tonkin Hill et al show some non-concordance and the rest of the studies seem to show neither. The list of transcript classes upregulated in DEGs across the studies in Fig 1b also seems to add little to the paper. Finally, in Fig 1c the authors present a selected set of genes of certain pathways and their expression levels with respect to parasite density. While the relationship is clear in some studies (Lee et al, Milner et al and Tonkin Hill et al) it is certainly not clear to me in the others.

The next section of the paper involves a reanalysis of a recent paper by the same group in which they showed that asymptomatic parasites persisting in the dry season circulated in the blood for longer (ie developed further into the asexual red cell cycle) than parasites from mild malaria cases in the wet season. By comparison with published in vitro development transcription profiles and by using a series of published techniques for estimating parasite age from transcription data they effectively reinforce the conclusion of their previous paper. I see that this in some ways lays the foundation for the meat of the paper that follows but since the informatic techniques that they utilise have already been validated and since they simply result in effectively the same (if slightly extended) conclusions as the original paper I am not sure that this section is really needed.

The remainder of the manuscript is devoted to providing a great deal of evidence that their conclusion (stated above) is valid and is thus the most important part. The authors first present heatmaps and associated timing of expression plots of DEGs in low vs high parasitaemia or non-severe vs severe disease according to the established in vitro gene expression timing. These analyses reinforce the idea of early gene expression patterns in high parasitaemia/severe cases and later gene expression patterns in low parasitaemia/non-severe cases for the data from Milner et al, Yamagishi et al and Lee et al but not for Tonkin-Hill et al (which is repeatedly misspelt in the paper). The authors explain the latter by the fact that the parasitaemia differences across severe/non-severe cases was low in this study. They go on to show no difference in samples comparing high and low transmission areas and use this as a type of negative control for the study. They next reanalyse the data from Daily et al. This paper purported to have discovered three novel transcriptional states from in-vivo parasites that were distinct from the in-vitro profile. By comparing transcription timing between these three states the authors do discover marked differences and equate these to clusters containing the highest parasitaemia or the most severe cases. However, the veracity of the conclusions of the Daily paper have been questioned (doi.org/10.1073/pnas.0811829106) and even if they were correct, the fact that they propose three novel transcriptional states would mean that it would be potentially problematic to use in-vitro expression timings for the analysis. It would have been much better in my view to analyse the data with respect to parasitaemia and if this was not available online, to request it from the

authors.

To further support their proposition, the authors then apply either of two previously published bioinformatic tools that estimate parasite age from transcription data. The first, a mixture model, shows significant parasite age differences in the predicted direction for the data from Milner et al, Yamagishi et al and Lee et al and this time for Tonkin-Hill et al. The second, a maximum likelihood estimate (MLE) finds significant age differences in data from Milner et al. and Tonkin-Hill et al but not for the other two. In figure 5a and b they use the same two informatic tools to show a negative relationship between either proportion of non-rings or parasite age and parasite density but only for the Milner et al and the Andrade et al study. They use data from the latter two studies to also show a negative relationship between parasite size and parasitaemia. They later repeat this analysis (Fig 5g) with the MLE method but add in two further datasets -Lee et al and Cordy et al. Thus, Fig 5b and g are thus effectively the same and don't need to be duplicated. Again, using the Milner et al and Andrade et al datasets they build a correlation matrix of variables from the two and demonstrate the predicted relationship. Finally, the authors use transcription data from an in-vivo infection of *P.coatneyi* in rhesus macaques to show that acute high parasitaemia is characterised by young rings whereas low chronic parasitaemia is dominated by older parasites. Overall, the authors have made a good case for their very interesting and novel conclusion that high parasitaemia tends to result in young ring stage parasites in the circulation whereas low parasitaemia is associate with older forms in the circulation. The use of multiple algorithms developed by others to determine parasite developmental stage from transcriptome data is to be congratulated. If the above hypothesis had not been possibly clouded by severe and non-severe cases, then the association may have been stronger. While there is clearly a relationship between severity and parasitaemia, this is complicated by differing epidemiological environments and patient recruitment strategies. For example, the Tonkin-Hill dataset where non-severe and severe cases overlapped significantly in parasitaemia, resulted in analyses where there were sometimes significant stage differences and in other analyses not. I suspect it would have been cleaner to analyse this dataset and the others on parasitaemia alone.

As mentioned above I have some issues with the analyses of the Daily et al data because of this paper's authors claim of novel transcriptional states would potentially confound the staging algorithms. The same issue applies to the Milner et al dataset where novel transcriptional states are also inferred but in this case segregating with parasitaemia. While I suspect that this may not be a confounder (particularly if these novel transcriptional states are an analytical artefact), there needs to be some discussion about it.

All in all, I think that the manuscript probably does present sufficient evidence to support the conclusion of an inverse relationship between parasitaemia and parasite age. It could be considerably shortened by omitting the analyses in Fig 1 and significantly shortening the discussion and data around Fig 2 (the authors previous data on which they had already made a similar conclusion). The other factor lacking is a more extensive discussion as to the possible mechanisms underlying the phenomenon described. It is not clear why antibodies should be present particularly to more adhesive parasites unless the hypothesis is that the antibodies do not recognise non-adhesive infected cells because the level of expression of adhesins is markedly decreased. Also, it has been shown that febrile temperatures in-vitro result in the adhesion of much younger parasites to endothelial receptors (doi.org/10.1073/pnas.1723989999) so although there will not be a clear relationship between parasitaemia and extent of fever, it is a possible confounder.

Reviewer #2 (Remarks to the Author):

Review of "Hidden transcriptional signatures of *P. falciparum* in different clinical presentations of malaria"

In this paper the authors have provided a follow-up meta-analysis to their recent work (Andrade et

al 2020, PMID: 33106664) which showed that asymptomatic malaria cases from the dry season in Mali had parasites from a larger developmental window in circulation compared to the wet season symptomatic cases. In the current paper, the authors have provided an in-depth analysis of transcriptomic data from Andrade et al, as well as 10 different studies that assessed differential expression patterns to a variety of conditions including disease severity, parasitaemia, and different transmission intensities. They illustrate how there is very little overlap across all of the studies but there are similar patterns of DEGs that correspond with high parasitaemia and disease severity between studies. They hypothesize that this is driven by differences in developmental stages present in circulation and go on to test this by predicting the stage composition based on how the differentially expressed genes identified are expressed over the IDC. They show that across studies, high parasitaemia and disease severity corresponds with only early stages of the parasite in circulation and the authors hypothesize that this may be driven by differences in adhesiveness in iRBCs. This work confirms what the authors had shown in Andrade et al 2020 and expands it to more broadly understand patterns of circulation in asymp/sypr cases and levels of parasitaemia.

I think this work, in combination with Andrade et al, represent a paradigm shift in how we think about parasite dynamics across the spectrum of disease severity in malaria. This paper will be of great interest to the malaria community as well as the wider field of infectious disease. My comments below are few and do not require additional analysis or data collection prior to publication.

Major comments:

- The conclusion that the observed differences in what is in circulation may be driven by differences in cytoadhesion needs more of a discussion of alternatives to this hypothesis. The evidence so far for this is based on differences in proportions of these stages in circulation, so it is unclear to me how you can differentiate between having greater cytoadherence (hiding in tissue) versus clearance (immune response/ phagocytosis targeting later stages). It seems possible that you could have the same number of later trophs in your samples, but with greater numbers of rings in the severe/high parasitaemia cases. How are you able to differentiate between these different scenarios?
- Additionally, along the cytoadherence conclusion, there is no analysis of cytoadhesion gene expression. You mention that it is difficult because it is so variable, but I think it would be really interesting if, controlling for differences in development, there were vars or other potential cytoadhesion genes that were DE depending on parasitaemia/symptoms.

Minor comments:

- Figures are hard to read in several places because of very small font.
- Line 572: the periodogram approach, can you additionally cite the methods paper as well as the Painter paper, please?
- Can the large supplementary tables be supplementary excel files? Very hard to follow or get info from in pdf format.

Reviewer #3 (Remarks to the Author):

In this article, the authors propose a model according to which the binding of Plasmodium falciparum-infected erythrocytes to vascular endothelial cells increases the rate of parasite growth, resulting in increased parasite burden and severity of disease presentation. Accordingly, circulating parasites in severe disease will be enriched for early stages of the intra-erythrocytic developmental cycle (IDC) relative to asymptomatic disease. Conversely, in asymptomatic disease, unsequestered parasites will be the norm, and these infections will be relatively enriched for parasites in later stages of the IDC. This work builds on a previous study by the team where they reported that

parasites from asymptomatic infections, prevalent in the dry season, differ in gene expression from those in symptomatic malaria cases observed in the wet season. The authors analyze transcription data from several studies that compare severe vs. mild/uncomplicated/asymptomatic (and high vs. low parasitemia) and find that the pattern of differentially expressed genes is consistent with this model, in that the genes with found to be significantly more expressed in severe disease/high parasitemia relative to mild/asymptomatic infection/low parasitemia are those found to be expressed early on in the IDC, in vitro. Conversely, genes found to be upregulated in mild relative to severe disease are those typically expressed later in the IDC, in vitro.

The authors present an interesting and compelling case that different expression patterns are associated with different levels of disease severity and of parasite burden, but not when a different variable (transmission intensity) is considered. Their case is strengthened by consistent results across studies, across methods to quantify transcription intensity (both microarray and RNA-seq), across geographic regions, etc. Overall the article is clear and well written.

However, the authors seem to overreach in their claim that it is parasite sequestration that leads to an increase in parasite burden. Do they envision a mechanism by which the former could cause the latter?

In fact, the observations are also consistent with other scenarios, including that it is the increase in parasite burden that leads to an increase in cytoadherence (possibly through some quorum sensing-like mechanism, which has been described in fungi) or that the two are independent of each other but dependent on other variable(s), such as host cues, genotype-dependent host exposure, or additional parasite factors.

Point by point reply to reviewers' comments on NCOMMS-21-09067 "Hidden transcriptional signatures of P. falciparum in different clinical presentations of malaria".

Reviewer #1:

This manuscript by Thomson-Luque et al undertakes a detailed analysis of a number of previous studies of gene transcription in P.falciparum field isolates. They conclude that parasites taken from patients with low parasitaemia or non-severe disease show a pattern of transcription consistent with later stages of the asexual red cell cycle than those taken from patients with either high parasitaemia or severe disease.

The first analysis that they undertake is to compare lists of differentially expressed genes DEGs in the different categories (high/low parasitaemia, severe/non severe disease) between the previous studies. I found this to be the weakest section of the paper. The circos plot in Figure 1a shows groups of DEGs from the various studies that are concordant between the studies. In fact, the only significant concordance by eye is between the studies of Andrade et al and Milner et al. The studies of Lee et al and Tonkin Hill et al show some non-concordance and the rest of the studies seem to show neither. The list of transcript classes upregulated in DEGs across the studies in Fig 1b also seems to add little to the paper. Finally, in Fig 1c the authors present a selected set of genes of certain pathways and their expression levels with respect to parasite density. While the relationship is clear in some studies (Lee et al, Milner et al and Tonkin Hill et al) it is certainly not clear to me in the others.

Our aim with Figure 1 is to set the stage with what the different studies had previously reported, and point out the limited or inconsistent concordance of DEGs across them, which the reviewer also remarks can be seen in Fig 1a. We have now attempted to make this message clearer in lines 108-109 of the revised manuscript. Likewise, in Fig. 1b we highlight that DEGs identified across the different studies did not result in enrichment of common terms following gene ontology or functional category analyses. And with Fig. 1c we show sets of genes of a few pathways identified in one or multiple studies in Fig. 1b and previously in Andrade et al., and we introduce for the first time in the manuscript the concept of comparing samples based on parasite load, and not necessarily based on proposed grouping by the authors in their original reports. We agree with the reviewer that the associations shown are not perfect, but we believe it does evidence trends across some of the studies that maybe of interest to the readers. We agree with the reviewer that Fig 1 is not the strongest point of our manuscript, but nevertheless highlights observations present on the original papers used in our reanalyses, and representis initial trends prompting us to investigate further as we did in subsequent figures.

The next section of the paper involves a reanalysis of a recent paper by the same group in which they showed that asymptomatic parasites persisting in the dry season circulated in the blood for longer (ie developed further into the asexual red cell cycle) than parasites from mild malaria cases in the wet season. By comparison with published in vitro development transcription profiles and by using a series of published techniques for estimating parasite age from transcription data they effectively reinforce the conclusion of their previous paper. I see that this in some ways lays the foundation for the meat of the paper that follows but since the informatic techniques that they utilise have already been validated and since they simply result in effectively the same (if slightly extended) conclusions as the original paper I am not sure that this section is really needed.

As the reviewer indicates, we have indeed previously shown that circulation time of infected erythrocytes is associated with major transcriptional differences that were linked to the developmental stage of circulating parasite. However, here we add a quantitative characterization of those findings with a series of already established bioinformatic approaches to set the analytical framework to apply to rest of the studies. We use the reanalysis of Andrade et al. almost a "positive control" to tell the readers (particularly the ones less acquainted with the bioinformatic tools) that our analytical framework indeed allows us to detect developmental differences between more and less developed parasites. Furthermore, the results obtained within this section are later used on the correlations with parasite size obtained through microscopy shown in Fig 5. We consider, as the reviewer suggests that this section lays the foundation and is relevant in validating much of the subsequent analyses.

The remainder of the manuscript is devoted to providing a great deal of evidence that their conclusion (stated above) is valid and is thus the most important part. The authors first present heatmaps and associated timing of expression plots of DEGs in low vs high parasitaemia or non-severe vs severe disease according to the established in vitro gene expression timing. These analyses reinforce the idea of early gene expression patterns in high parasitaemia/severe cases and later gene expression patterns in low parasitaemia/non-severe cases for the data from Milner et al, Yamagishi et al and Lee et al but not for Tonkin-Hill et al (which is repeatedly misspelt in the paper). The authors explain the latter by the fact that the parasitaemia differences across severe/non-severe cases was low in this study. They go on to show no difference in samples comparing high and low transmission areas and use this as a type of negative control for the study.

We thank the reviewer for the comment, and apologize for misspelling Tonkin-Hill. We have now corrected it on lines 237 and 246 of the revised manuscript.

They next reanalyse the data from Daily et al. This paper purported to have discovered three novel transcriptional states from in-vivo parasites that were distinct from the in-vitro profile. By comparing transcription timing between these three states the authors do discover marked differences and equate these to clusters containing the highest parasitaemia or the most severe cases. However, the veracity of the conclusions of the Daily paper have been questioned (doi.org/10.1073/pnas.0811829106) and even if they were correct, the fact that they propose three novel transcriptional states would mean that it would be potentially problematic to use in-vitro expression timings for the analysis. It would have been much better in my view to analyse the data with respect to parasitaemia and if this was not available online, to request it from the authors.

We agree with the reviewer that the three the clusters reported in Daily et al may not truly represent distinct parasite transcriptional states. The three groups of samples resulted from the non-negative matrix factorization (NMF) method employed, and we are aware that these results have been questioned by Lemieux et al and others; and that differences in content of sexual stages of samples in the different clusters has been suggested to promote the transcriptional changes. Nevertheless, NMF clustering is widely employed and has led to segregation aligned with parasitemia in other studies, such as Milner et al.. We have edited the manuscript in lines 70 and 96 to refer to this study not as comparing distinct parasite physiological states, but instead as comparing NMF-generated clusters of samples.

We also agree with the reviewer that would be nice to have full knowledge of parasite density of samples included in Daily et al. to run the reanalysis, but these were not provided in the supplementary data nor in the raw data of the manuscript. Still, we were encouraged to see that our results were in accordance with what Daily and colleagues report regarding cluster 2 comprising the highest average parasitaemia and closest to the early ring-stage transcriptional profile.

The purpose of our manuscript was not to question the conclusions of any of the previous reports, but to investigate if different parasite developmental stage-associated transcriptional patterns align with our hypothesis. We hope that our manuscript will lead authors from previous, and especially future studies to investigate their data considering the proposed relation between parasite density and time of circulation and that it helps correcting some of the misconceptions potentially promoted by previous analyses.

To further support their proposition, the authors then apply either of two previously published bioinformatic tools that estimate parasite age from transcription data. The first, a mixture model, shows significant parasite age differences in the predicted direction for the data from Milner et al, Yamagishi et al and Lee et al and this time for Tonkin-Hill et al. The second, a maximum likelihood estimate (MLE) finds significant age differences in data from Milner et al. and Tonkin-Hill et al but not for the other two.

We agree with the reviewer that the results from the individual analyses do not fit perfectly to all studies but we believe that all approaches together make our results very robust, especially knowing that the different studies use different sample sizes, age of participants, sequencing techniques and methodologies to calculate DEGs.

In figure 5a and b they use the same two informatic tools to show a negative relationship between either proportion of non-rings or parasite age and parasite density but only for the Milner et al and the Andrade et al study. They use data from the latter two studies to also show a negative relationship between parasite size and parasitaemia. They later repeat this analysis (Fig 5g) with the MLE method but add in two further datasets -Lee et al and Cordy et al. Thus, Fig 5b and g are thus effectively the same and don't need to be duplicated.

The reviewer is correct that data shown in Fig. 5b was also included in Fig. 5g. We had done so, for easiness of telling the story of the two studies for which we had also microscopy data later. But in response to the reviewer's comment, we have now altered Fig. 5a and b to include data from three and four studies respectively (Andrade et al., Milner et al. and Lee et al. in Fig. 5a, and plus Cordy et al. in Fig. 5b) and have hence deleted Fig. 5g. The alterations are highlighted in lines 344-348 of the revised manuscript.

Again, using the Milner et al and Andrade et al datasets they build a correlation matrix of variables from the two and demonstrate the predicted relationship. Finally, the authors use transcription data from an in-vivo infection of *P. coatneyi* in rhesus macaques to show that acute high parasitaemia is characterised by young rings whereas low chronic parasitaemia is dominated by older parasites.

Overall, the authors have made a good case for their very interesting and novel conclusion that high parasitaemia tends to result in young ring stage parasites in the circulation whereas low parasitaemia is associate with older forms in the circulation. The use of multiple algorithms developed by others to determine parasite developmental stage from transcriptome data is to be congratulated. If the above hypothesis had not been possibly clouded by severe and non-severe cases, then the association may have been stronger. While there is clearly a relationship between severity and parasitaemia, this is complicated by differing epidemiological environments and patient recruitment strategies. For example, the Tonkin-Hill dataset where non-severe and severe cases overlapped significantly in parasitaemia, resulted in analyses where there were sometimes significant stage differences and in other analyses not. I suspect it would have been cleaner to analyse this dataset and the others on parasitaemia alone.

We thank the reviewer for the comment, and agree that segregation based parasitaemia only, and possibly similar technical and analytical methods to calculate expression levels and DEGs across the different studies of could have made associations even stronger, as we now point in line 428 of the revised manuscript.

As mentioned above I have some issues with the analyses of the Daily et al data because of this paper's authors claim of novel transcriptional states would potentially confound the staging algorithms. The same issue applies to the Milner et al dataset where novel transcriptional states are also inferred but in this case segregating with parasitaemia. While I suspect that this may not be a confounder (particularly if these novel transcriptional states are an analytical artefact), there needs to be some discussion about it.

As the reviewer, we also suspect that Daily and Milner clustering of samples is likely not a major confounder, particularly in the case of Milner et al because the clustering segregated samples aligned with parasitaemia differences. Furthermore, our reanalysis interpretation addresses only the few lines in the original paper referring to parasite densities and predicted stages within the groups, and we do not discuss nor mention the physiological states proposed at the time. As mentioned above we have edited the manuscript in lines 70 and 97 to refer to this study not as comparing distinct parasite physiological states, but instead as comparing NMF-generated clusters of samples.

All in all, I think that the manuscript probably does present sufficient evidence to support the conclusion of an inverse relationship between parasitaemia and parasite age. It could be considerably shortened by omitting the analyses in Fig 1 and significantly shortening the discussion and data around Fig 2 (the authors previous data on which they had already made a similar conclusion).

We thank the reviewer for the comment. As mentioned above, we believe that outlining how the previously existing data from the multiple studies partially intersected as we do in Fig 1 may be important for some of the readership of our manuscript. Also, we think that providing a detailed description of our approach on the set of samples where the differences in parasite circulation time were first described and previously published, as we do in Fig. 2, supports the rest of our manuscript and its conclusions.

The other factor lacking is a more extensive discussion as to the possible mechanisms underlying the phenomenon described. It is not clear why antibodies should be present particularly to more adhesive parasites unless the hypothesis is that the antibodies do not recognise non-adhesive infected cells because the level of expression of adhesins is markedly decreased. Also, it has been shown that febrile temperatures in-vitro result in the adhesion of much younger parasites to endothelial receptors () so although there will not be a clear relationship between parasitaemia and extent of fever, it is a possible confounder.

The molecular mechanism underlying the longer time in circulation of parasites in low parasitemia/decreased severity cases likely relies on different binding capacities of infected erythrocytes to the vascular endothelium but remains for now unknown. We hypothesize that sequential expression of parasite receptors on the surface infected erythrocytes starting from stronger and progressing to weaker binders in response to humoral immunity could lead to parasitaemias so low that an efficient antibody response would not be triggered. However, we do not know if this could happen through decreased expression of the surface molecules, or through normal expression levels of particular types of surface molecules with poor binding ability. Nevertheless, we agree with the reviewer that alternative and potentially non-exclusive mechanisms may be at play. We have now added to the discussion of the revised manuscript (lines 486-492) the hypotheses that cytokine environment associated with febrile temperatures in clinical malaria cases and the increased temperature itself may also affect endothelial binding.

Reviewer #2:

In this paper the authors have provided a follow-up meta-analysis to their recent work (Andrade et al 2020, PMID: 33106664) which showed that asymptomatic malaria cases from the dry season in Mali had parasites from a larger developmental window in circulation compared to the wet season symptomatic cases. In the current paper, the authors have provided an in-depth analysis of transcriptomic data from Andrade et al, as well as 10 different studies that assessed differential expression patterns to a variety of conditions including disease severity, parasitaemia, and different transmission intensities. They illustrate how there is very little overlap across all of the studies but there are similar patterns of DEGs that correspond with high parasitaemia and disease severity between studies. They hypothesize that this is driven by differences in developmental stages present in circulation and go on to test this by predicting the stage composition based on how the differentially expressed genes identified are expressed over the IDC. They show that across studies, high parasitaemia and disease severity corresponds with only early stages of the parasite in circulation and the authors hypothesize that this may be driven by differences in adhesiveness in iRBCs. This work confirms what the authors had shown in Andrade et al 2020 and expands it to more broadly understand patterns of circulation in asymp/symp cases and levels of parasitaemia.

I think this work, in combination with Andrade et al, represent a paradigm shift in how we think about parasite

dynamics across the spectrum of disease severity in malaria. This paper will be of great interest to the malaria community as well as the wider field of infectious disease. My comments below are few and do not require additional analysis or data collection prior to publication.

We thank the reviewer for the comment, and we share the reviewer's view that this manuscript may aid in further understanding the factors characterizing the development of the whole spectrum of malaria clinical presentations.

Major comments:

- The conclusion that the observed differences in what is in circulation may be driven by differences in cytoadhesion needs more of a discussion of alternatives to this hypothesis. The evidence so far for this is based on differences in proportions of these stages in circulation, so it is unclear to me how you can differentiate between having greater cytoadherence (hiding in tissue) versus clearance (immune response/ phagocytosis targeting later stages). It seems possible that you could have the same number of later trophs in your samples, but with greater numbers of rings in the severe/high parasitaemia cases. How are you able to differentiate between these different scenarios?

We agree with the reviewer that there are still many unknowns, and we also point in the manuscript introduction and discussion that immunity is surely central for protection against clinical malaria in endemic areas. We can envision cases where parasites cytoadhere efficiently and elicit strong immune response promoting phagocytosis of later stage parasites. What we suggest, is that following efficient humoral responses there should be clearance of that particular variant and the next variants will progressively be less efficient binders and hence grow a little bit less well. So, immunity is helped by slower growers in keeping low parasitaemias, until the situation we describe in Andrade et al. is achieved: very low parasitaemias and a barely active immune response. We are likely unable to differentiate between the scenarios the reviewer proposes, but we think parasites heavily target with immune response/ phagocytosis towards later stages will very quickly produce few rings and hence have lower parasitaemias, or they must switch to a less immunogenic variant and again the equilibrium between the stages found in circulation can be driven by sequestration ability. We added a few lines to make the interpretation of our analyses more open (lines 486-492 of the revised manuscript) hoping future studies will help clarify unsolved questions, and that the discussion we started broadens to the larger malaria research community.

- Additionally, along the cytoadherence conclusion, there is no analysis of cytoadherence gene expression. You mention that it is difficult because it is so variable, but I think it would be really interesting if, controlling for differences in development, there were vars or other potential cytoadhesion genes that were DE depending on parasitaemia/symptoms.

Within the 10 studies we reanalysed, only Andrade et al. and Tonkin-Hill et al discuss the association between disease severity and PfEMP1 transcripts. In Andrade et al. we did not see statistically significant enrichment of particular vars, and although we saw a trend for higher expression of the top expressed var genes in individuals with clinical malaria vs asymptomatic in the dry season, this was not statistically significant. Tonkin-Hill et al. with a developed de novo assembly pipeline of RNAseq data also analysed var gene expression and detected no difference between severe malaria and uncomplicated malaria in the total number of var gene reads. However segregation at the multidomain and individual domain level between severe and non-severe disease was identified. Tonkin-Hill et al. also found genes involved in PfEMP1 transport and regulation to be down-regulated in severe malaria leading the authors to suggest that var gene expression was reduced, which we believe could also highlight less developed stage of parasites in these severe cases. We added lines 477-485 of the revised manuscript to discuss this important point.

Nevertheless, we agree that it is of great interest to investigate parasite cytoadherence over the malaria clinical spectrum, and indeed, in a separate set of ongoing longitudinal experiments, we are comparing expression of cytoadhesion-related genes of developmental stage-matched single cell parasites collected from subjects at several points during the dry season versus subjects with diverse acute symptomatic malaria during the rainy season. However, this is a different set of experiments and analyses which are just beginning and may span several malaria seasons, and therefore we hope the reviewer might appreciate that these additional experiments, would be beyond the scope of the current manuscript.

Minor comments:

- Figures are hard to read in several places because of very small font.

We apologize for the small version of some of the figures included in the text. We have in the revised manuscript made also all figures full size and outside of the text.

- Line 572: the periodogram approach, can you additionally cite the methods paper as well as the Painter paper, please?

We now cite Painter et al alongside the original methods paper by Glynn and colleagues in 2006, as highlighted on line 586 of the revised manuscript.

• Can the large supplementary tables be supplementary excel files? Very hard to follow or get info from in pdf format.

We agree with the reviewer that pdf format makes tables very hard to follow, and we now also provide all supplementary tables as excel files.

Reviewer #3:

In this article, the authors propose a model according to which the binding of Plasmodium falciparum-infected erythrocytes to vascular endothelial cells increases the rate of parasite growth, resulting in increased parasite burden and severity of disease presentation. Accordingly, circulating parasites in severe disease will be enriched for early stages of the intra-erythrocytic developmental cycle (IDC) relative to asymptomatic disease. Conversely, in asymptomatic disease, unsequestered parasites will be the norm, and these infections will be relatively enriched for parasites in later stages of the IDC. This work builds on a previous study by the team where they reported that parasites from asymptomatic infections, prevalent in the dry season, differ in gene expression from those in symptomatic malaria cases observed in the wet season. The authors analyze transcription data from several studies that compare severe vs. mild/uncomplicated/asymptomatic (and high vs. low parasitemia) and find that the pattern of differentially expressed genes is consistent with this model, in that the genes with found to be significantly more expressed in severe disease/high parasitemia relative to mild/asymptomatic infection/low parasitemia are those found to be expressed early on in the IDC, in vitro. Conversely, genes found to be upregulated in mild relative to severe disease are those typically expressed later in the IDC, in vitro.

The authors present an interesting and compelling case that different expression patterns are associated with different levels of disease severity and of parasite burden, but not when a different variable (transmission intensity) is considered. Their case is strengthened by consistent results across studies, across methods to quantify transcription intensity (both microarray and RNA-seq), across geographic regions, etc. Overall the article is clear and well written.

We thank the reviewer for the comment.

However, the authors seem to overreach in their claim that it is parasite sequestration that leads to an increase in parasite burden. Do they envision a mechanism by which the former could cause the latter?

In fact, the observations are also consistent with other scenarios, including that it is the increase in parasite burden that leads to an increase in cytoadherence (possibly through some quorum sensing-like mechanism, which has been described in fungi) or that the two are independent of each other but dependent on other variable(s), such as host cues, genotype-dependent host exposure, or additional parasite factors.

A mechanism where high parasite burdens lead to increased sequestration is indeed possible, cytokines such as TNF and IL-1 have been shown to increase expression of ligands on endothelial cells which may promote adhesion of infected cells. Also, fever induced by high parasitaemias has been suggested to affect adhesion of infected erythrocytes. However, in our view such mechanism will constantly lead to increased circulation of higher parasitaemias of young parasite stages, culminating in the patients' death or effective immune response against the dominant variant, forcing the parasite to switch sequentially until parasitaemia is controlled and eventually cleared. What we suggest is that the parasite stages found circulating in each patient in the different studies may indicate the maximum parasite growth in the given host immune context.

The reviewer is likely correct that we overreached in our claims, and we have now attempted to keep the discussion more open to alternative possibilities (lines 486-492 of the revised manuscript), and we hope studies following ours will help clarify these questions.

Reviewer comments, second round –

Reviewer #1 (Remarks to the Author):

The authors have addressed many of my previous comments, however there a number that they have not and which I believe still need attention.

1) Figure 1 is in my view still the weakest point in the paper and should be moved to supplementary material. The data displayed, as I commented before, are not totally consistent with the arguments presented.

2) I still believe that the second section that reanalyses the authors previous work and comes to the same conclusion as the original paper, while an important instance of applying the staging algorithms used later, could be shortened.

3) The authors persist with strongly defending their hypothesis that the antibody response to the infected red cell surface selects over time for parasites with decreased adhesion such that in semi-immune asymptomatic individuals, circulating parasites are less adhesive and therefore more susceptible to splenic clearance. As far as I know there is no evidence for antibodies being preferentially directed at more adhesive Pfemp1, rifin or stevor proteins and if there were, I would assume that the authors would have quoted it. I think that the discussion should therefore be more open to other interpretations for which there is at least some evidence (fever increases adhesion, lower var expression at low parasitaemia hinted at in the authors previous paper and in [doi:10.1371/journal.pone.0114401](https://doi.org/10.1371/journal.pone.0114401)). A more balanced evaluation of the possible mechanisms would I think be a more fitting ending to what is a potentially very important paper.

Reviewer #2 (Remarks to the Author):

The reviewers have sufficiently addressed all of my comments.

Point by point reply to Reviewers' Comments

Reviewer #1 (Remarks to the Author):

The authors have addressed many of my previous comments, however there a number that they have not and which I believe still need attention.

1) Figure 1 is in my view still the weakest point in the paper and should be moved to supplementary material. The data displayed, as I commented before, are not totally consistent with the arguments presented.

We attempted to make our interpretation of the data more open, and toned down the last sentence of the figure description on lines 127-129 and edited also a few words highlighted in that paragraph.

2) I still believe that the second section that reanalyses the authors previous work and comes to the same conclusion as the original paper, while an important instance of applying the staging algorithms used later, could be shortened.

We have shortened a few words on a few sentences of this section, and we believe that the consistency of the data presented, although reading a bit repetitive, shows the strength of the approach that will be used in later figures of the paper, which we think is important for the broad readership that we hope to reach with our manuscript.

3) The authors persist with strongly defending their hypothesis that the antibody response to the infected red cell surface selects over time for parasites with decreased adhesion such that in semi-immune asymptomatic individuals, circulating parasites are less adhesive and therefore more susceptible to splenic clearance. As far as I know there is no evidence for antibodies being preferentially directed at more adhesive PfEMP1, rifin or stevor proteins and if there were, I would assume that the authors would have quoted it. I think that the discussion should therefore be more open to other interpretations for which there is at least some evidence (fever increases adhesion, lower var expression at low parasitaemia hinted at in the authors previous paper and in doi:10.1371/journal.pone.0114401). A more balanced evaluation of the possible mechanisms would I think be a more fitting ending to what is a potentially very important paper.

The reviewer is correct that reports showing hierarchical acquisition of anti-PfEMP1s would strengthen our proposed hypothesis, and we apologise for not including those earlier. We now mention data from studies in Mali and Tanzania showing that antibodies against PfEMP1 domains of A and A/B types, or domains binding to EPCR are acquired earlier in life compared to others, and added lines 315-317 and the two references. Also, we now include on lines 385-386, the possibility of being the reduced PfEMP1 expression, independent of the expressed var type, that leads to the longer circulation of iRBCs. Additionally, we also include a recent report showing that complement component 1s (C1s) in serum cleaves PfEMP1 and reduces cytoadherence which could be another way to affect cytoadhesion without antibodies involved. We agree that our discussion is not perfectly balanced on the different hypotheses, but we are clear that they are all hypotheses and all worth of investigation.